# Synchronized HIV assembly by tunable PIP$_2$ changes reveals PIP$_2$ requirement for stable Gag anchoring

Frauke Mücksch[1], Vibor Laketa[1,2], Barbara Müller[1], Carsten Schultz[3,4], Hans-Georg Kräusslich[1,2]*

[1]Department of Infectious Diseases, Virology, University Hospital Heidelberg, Heidelberg, Germany; [2]German Center for Infectious Disease Research, Partner site Heidelberg, Braunschweig, Germany; [3]Cell Biology and Biophysics Unit, European Molecular Biology Laboratory, Heidelberg, Germany; [4]Department of Physiology and Pharmacology, Oregon Health and Science University, Portland, United States

**Abstract** HIV-1 assembles at the plasma membrane (PM) of infected cells. PM association of the main structural protein Gag depends on its myristoylated MA domain and PM PI(4,5)P$_2$. Using a novel chemical biology tool that allows rapidly tunable manipulation of PI(4,5)P$_2$ levels in living cells, we show that depletion of PI(4,5)P$_2$ completely prevents Gag PM targeting and assembly site formation. Unexpectedly, PI(4,5)P$_2$ depletion also caused loss of pre-assembled Gag lattices from the PM. Subsequent restoration of PM PI(4,5)P$_2$ reinduced assembly site formation even in the absence of new protein synthesis, indicating that the dissociated Gag molecules remained assembly competent. These results reveal an important role of PI(4,5)P$_2$ for HIV-1 morphogenesis beyond Gag recruitment to the PM and suggest a dynamic equilibrium of Gag-lipid interactions. Furthermore, they establish an experimental system that permits synchronized induction of HIV-1 assembly leading to induced production of infectious virions by targeted modulation of Gag PM targeting.

*For correspondence: hans-georg.kraeusslich@med.uni-heidelberg.de

## Introduction

Human immunodeficiency virus type 1 (HIV-1) particles assemble and bud at the plasma membrane (PM) of infected cells. This requires trafficking of the main structural component Gag from its site of synthesis at cytosolic polysomes to the inner leaflet of the PM. Here, Gag assembles into a multi-meric lattice comprising ~2500 Gag molecules. Gag also recruits the other constituents of the infectious virion as well as components of the cellular endosomal sorting complex required for transport (ESCRT) to nascent budding sites (reviewed in *Sundquist and Kräusslich, 2012*).

Gag consists of the individually folded MA (matrix), CA (capsid), NC (nucleocapsid) and p6 domains, which separate upon proteolytic maturation to the infectious virion. CA and NC mediate protein and RNA interactions during virion morphogenesis, while p6 recruits ESCRT components. PM targeting of Gag depends on the N-terminal MA domain. Mutational analyses revealed that the N-terminal myristoyl moiety and a surface exposed patch of basic residues in MA (the highly basic region, HBR) are important for Gag targeting, membrane association and virus formation (*Lorizate and Kräusslich, 2011*; *Freed, 2015*). However, these two features would not be sufficient to explain specific PM targeting. The phosphoinositide phosphatidylinositol 4,5-bisphosphate (PI(4,5)P$_2$) is known to act as a cue for specific recruitment of many peripheral PM proteins and was also shown to be important for Gag PM targeting. Depleting PI(4,5)P$_2$ abolished PM targeting of Gag, while increasing PI(4,5)P$_2$ at intracellular membranes redirected Gag to those sites (*Ono et al.,*

**eLife digest** Viruses are parasites that must infect the living cell of a host in order to grow and replicate. To do so, the virus attaches to the host's cell and transfers its genetic material to the inside. The virus then hijacks the cell and forces it to build proteins and more genetic material that will assemble into new copies of the virus, and the completed virus particles are released to infect new cells.

The HIV-1 virus encodes a protein called Gag that coordinates the assembly of new copies of this virus. Thousands of Gag proteins accumulate into a lattice at the inside of the host cell membrane by an unknown mechanism, where they gather all essential components to form a new infectious virus. Once completed, the virus particle detaches itself from the host cell and Gag gives the virus its structure.

Previous research by several groups has shown that a lipid molecule found in cell membranes called PIP$_2$ can bind to Gag and helps the virus to assemble. However, until now, it was unclear if PIP$_2$ anchors Gag to the cell membrane of living cells and if it plays any other roles in the later stages of assembly.

Now, Mücksch et al. have developed a new approach to study this question by rapidly manipulating the levels of PIP$_2$ in living cells and monitoring Gag assembly through a fluorescent marker. The experiments showed that PIP$_2$ was needed to initiate the assembly process, but also to maintain the partially assembled Gag lattice at the membrane. When PIP$_2$ was removed, Gag proteins could not gather at the cell membrane and the already assembled Gag lattices broke down. When PIP$_2$ levels were increased again, the Gag proteins that had disappeared from the membrane formed new lattices. This suggests that PIP$_2$ has a much broader role in HIV-1 particle formation than previously assumed.

A next step will be to use this new experimental approach to study how particles of HIV are assembled and released. This knowledge may help scientists to develop antiviral drugs that interfere with the assembly step of the virus that could be used to prevent or treat HIV infections.

2004). Consistently, several studies reported that presence of PI(4,5)P$_2$ enhances binding of Gag-derived proteins to synthetic liposomes in vitro (*Chukkapalli et al., 2008*, *2010*; *Dick et al., 2012*; *Olety and Ono, 2014*; *Mercredi et al., 2016*), and PI(4,5)P$_2$ appeared to be enriched in the HIV-1 lipidome (*Chan et al., 2008*).

Gag seems to dominate in a monomeric form or in small oligomers in the cytoplasm (*Kutluay and Bieniasz, 2010*; *Hendrix et al., 2015*), and its N-terminal myristate group is proposed to be buried within the globular domain of MA at this stage (*Spearman et al., 1997*; *Saad et al., 2006*; *Tang et al., 2004*). Gag interaction with PI(4,5)P$_2$ has been suggested to trigger exposure of the myristate moiety followed by its insertion into the inner leaflet of the PM, thereby anchoring Gag. Recent models propose a regulatory role of RNA for this myristoyl switch (*Chukkapalli et al., 2010*; *Alfadhli et al., 2011*; *Chukkapalli et al., 2013*; *Kutluay et al., 2014*). In the cytosol, specific cellular tRNAs are bound to the basic region of MA in the absence of PI(4,5)P$_2$, thus preventing premature Gag association with intracellular membranes. This MA-tRNA interaction may be outcompeted by PI(4,5)P$_2$ at the PM, thereby triggering the myristoyl switch.

The NMR structure of HIV-1 MA bound to a PI(4,5)P$_2$ molecule carrying truncated acyl chains revealed the 2′ acyl chain to be buried in the myristoyl-binding pocket of MA (*Saad et al., 2006*). This observation suggested a model for Gag membrane anchoring, which was subsequently challenged by a recent report from the same group, however (*Mercredi et al., 2016*). Upon arrival of Gag at the PM, the 2′ acyl chain of PI(4,5)P$_2$ may be flipped outward and displace myristate from the acyl binding pocket of MA. Myristate in turn would insert into the PM. The consequent exchange of the unsaturated 2′ acyl chain of PI(4,5)P$_2$ by the saturated myristic acid would increase local lipid saturation and membrane order. If occurring in all or a majority of the ~2500 tightly packed Gag molecules, this mechanism could also explain the observed liquid-ordered state of the HIV envelope (*Lorizate et al., 2009*; *Brügger et al., 2006*; *Chan et al., 2008*). This model would further suggest

that Gag clusters are stably anchored to the PM *via* their myristoyl moiety and PI(4,5)P$_2$ would then be expected to be dispensable once anchoring has occurred.

PI(4,5)P$_2$ is clearly important for Gag PM targeting, but the dynamics of PI(4,5)P$_2$ Gag interaction and the role of PI(4,5)P$_2$ in later stages of HIV assembly have not been examined so far. To do this, we need to monitor Gag localization in real time while manipulating PI(4,5)P$_2$ levels in living cells. Here, we made use of a recently developed reversible chemical dimerizer system (abbreviated rCDS) allowing rapid and controlled depletion and reconstitution of PM PI(4,5)P$_2$ levels in living, virus-producing cells by a small molecule (*Feng et al., 2014*; *Schifferer et al., 2015*). Using this system, we showed that the nascent HIV-1 Gag assembly site is highly dependent on PI(4,5)P$_2$ during the entire assembly process, and membrane association of apparently complete assembly sites remained PI(4,5)P$_2$ dependent. PI(4,5)P$_2$ removal from the PM not only abolished Gag PM targeting, but also caused dissociation of pre-assembled Gag clusters from the membrane. Reconstitution of *bona fide* Gag assembly sites at the PM was observed upon re-establishment of PM PI(4,5)P$_2$ levels. Our findings are inconsistent with stable anchoring of large Gag clusters at the PM through myristoyl moieties alone and suggest a highly dynamic mode of Gag PM binding. Furthermore, this approach

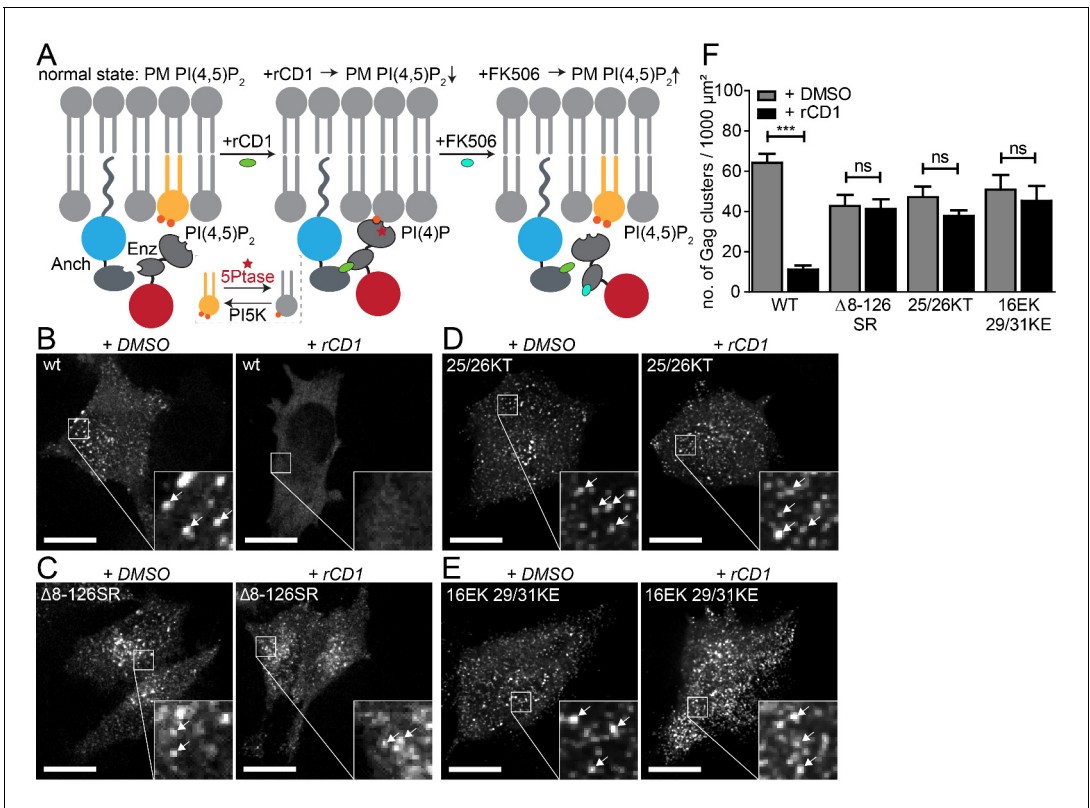

**Figure 1.** PI(4,5)P$_2$ is required for Gag assembly in living cells in an MA-dependent manner. (**A**) Principle of the reversible chemical dimerizer system (rCDS). Translocation of the Enzyme 5Ptase to the PM by the reversible chemical dimerizer rCD1 leads to breakdown of PI(4,5)P$_2$ to PI(4)P. Addition of the competing ligand FK506 displaces the enzyme from the PM, which restores PI(4,5)P$_2$ levels rapidly. See also *Figure 1—figure supplement 1* and *Video 1*, which show reversible PI(4,5)P2 depletion by the rCDS in living cells. (**B–E**) Representative SDC fluorescence images of the ventral PM of HeLa Kyoto cells transfected with the Anchor (LCK-ECFP-SNAP), Enzyme (mRFP-FKBP-5Ptase) and the indicated HIV-1 derived constructs. Cells were treated with 1% DMSO (left panels) or 1 μM rCD1 (right panels) at 4 hpt. Gag was detected at 22 hpt at the ventral membranes of cells via EGFP (in living cells, pCHIV derived constructs in (**B**) and (**C**)) or by indirect immunolabeling (in fixed cells, NL4-3 derived constructs in (**D**) and (**E**)). Arrows indicate Gag clusters. Scale bar represents 20 μm. (**F**) Number of Gag clusters detected at the ventral membrane per 1000 μm$^2$ membrane area. Error bars represent the standard error of the mean (statistical significance was assessed with the two-tailed unpaired Student's t-test; ***p≤0.001) of n = 27/26; 15/18; 35/33; 29/22 cells (from n = 3; 2; 2; 2 independent experiments), respectively.

The following figure supplement is available for figure 1:

**Figure supplement 1.** Reversible PI(4,5)P$_2$ depletion from the PM by the rCDS.

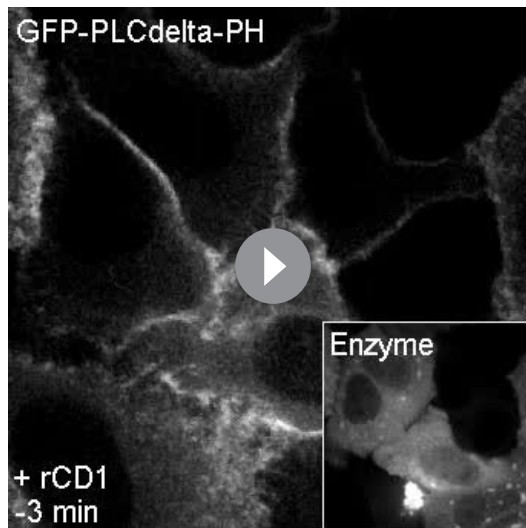

**Video 1.** Reversible PI(4,5)P$_2$ depletion from the PM by the rCDS. Representative time-lapse SDC fluorescence microscopy sequence of the central volume of HeLa Kyoto cells transfected with the plasmids expressing the rCDS and the PI(4,5)P$_2$ detector protein EGFP-PLCδ-PH. The maximum intensity projection of four focal planes acquired with an axial spacing of 0.5 μm is shown. Treatment with 1 μM rCD1 at 22 hpt induced translocation of the Enzyme to the PM (see insert box), leading to depletion of EGFP-PLCδ-PH from the PM. Subsequent addition of 1 μM FK506 releases the Enzyme from the PM (see insert box) and restored EGFP-PLCδ- PH binding. The sequence was acquired at a time resolution of 2 min/frame and is displayed with 6.2 fps. See also *Figure 1—figure supplement 1A* for corresponding video still images.

allows synchronization of assembly and release of infectious HIV-1. These processes are usually highly asynchronous, both within a cell population and on the level of individual cells (*Ivanchenko et al., 2009*).

## Results

The key parts of the reversible chemical dimerizer system (rCDS; *Figure 1A*) used in this study are a PM anchor (LCK-ECFP-SNAP, here referred to as 'Anchor'), and a cytosolic enzyme-bearing construct (FKBP-mRFP-5Ptase, here referred to as 'Enzyme'), which catalyzes the conversion of PI(4,5)P$_2$ to PI(4)P (*Figure 1A*) (*Schifferer et al., 2015*). Turnover of PM PI(4,5)P$_2$ is induced by addition of a cell-permeant small chemical dimerizer (rCD1) (*Feng et al., 2014*), which binds to both the SNAP and FKBP moieties of the respective fusion proteins and thereby links Anchor and Enzyme. Accordingly, rCD1 addition leads to rapid PM recruitment of the Enzyme and consequent depletion of PI(4,5)P$_2$ from the PM, while PI(4,5)P$_2$ levels at the PM remain unchanged in the absence of this compound. Under our conditions, PM recruitment of the phosphatase was complete within 5 min after rCD1 addition. PI(4,5)P$_2$ depletion can be monitored through dissociation from the PM of an EGFP-tagged pleckstrin homology (PH) domain of PLCδ; its dissociation was observed as early as 2 min after rCD1 addition in HeLa cells (*Figure 1—figure supplement 1A*, *Video 1*). PM recruitment of the Enzyme can be reversed by adding the rCD1 competitor FK506 (*Figure 1A*). FK506 outcompetes the binding of rCD1 to the FKBP domain causing removal of the Enzyme from the PM within seconds and replenishment of normal PI(4,5)P$_2$ levels by endogenous PI(4)P$_5$ kinases. Full re-localization of the PH-domain reporter to the PM was detected 2 min after addition of FK506, indicating rapid and efficient recovery of PI(4,5)P$_2$ (*Figure 1—figure supplement 1A*, *Video 1*).

To analyze the relevance of PI(4,5)P$_2$ for HIV-1 assembly site formation in transfected or infected cells, rCD1-mediated PI(4,5)P$_2$ depletion needs to be sustained for long periods (>12 hr), while maintaining the potential of rapid reversal and without unwanted toxic effects. These prerequisites were evaluated in HeLa cells co-transfected with plasmids expressing the rCDS components and the PI(4,5)P$_2$ indicator EGFP-PLCδ-PH. rCD1 was added 4 hr after transfection. Monitoring localization of the PI(4,5)P$_2$ reporter over time revealed complete loss of PM recruitment up to 22 hr post transfection without apparent cytotoxicity (*Figure 1—figure supplement 1B*, left). PM PI(4,5)P$_2$ levels were rapidly restored upon addition of FK506 at this time point (*Figure 1—figure supplement 1B*). We further observed that Gag assembly was not significantly affected by co-expression of Anchor and Enzyme (p=0.3714) (*Figure 1—figure supplement 1C and D*), thus validating that the rCDS is applicable to study Gag recruitment and assembly in living cells.

### PI(4,5)P$_2$ is required for Gag assembly at the PM of living cells in a MA-dependent manner

We first analyzed Gag assembly at the ventral PM in co-transfected HeLa cells that were subjected to PM PI(4,5)P$_2$ depletion through rCD1 prior to Gag accumulation. For visualization of assembly

sites in live cells, Gag tagged with EGFP was expressed in the viral context from a non-infectious subviral construct expressing all HIV-1 proteins except for Nef (equimolar mixture of pCHIV and pCHIV[EGFP] [*Lampe et al., 2007*]). Untagged Gag from proviral plasmids was detected in fixed cells by immunostaining. Confocal z-stacks covering the whole cell volume were acquired and analyzed in order to discriminate between assemblies at the ventral PM and punctuate signals arising from intra-cellular Gag clusters. Gag clusters at the PM corresponding to HIV-1 assembly sites were rarely observed when rCD1 was added at 4 hr after transfection (*Figure 1B*, right) whereas DMSO treated control cells showed a high number of Gag clusters (*Figure 1B*, left). Quantification of Gag clusters at the ventral PM of DMSO (n = 27) or rCD1 treated (n = 26) cells revealed a pronounced (80%) and highly significant (p<0.0001) reduction of Gag clusters in PM PI(4,5)P$_2$ depleted cells compared to control cells (*Figure 1F*).

Interaction of Gag with PI(4,5)P$_2$ in vitro is mediated by the N-terminal MA domain and mutations in MA affect Gag PM localization. In order to gain further insight into the role of MA for the observed PI(4,5)P$_2$-dependent Gag recruitment, we employed a Gag variant that retains the myristoylation signal and C-terminal cleavage site, but carries a deletion of the entire globular MA domain (Δ8–126 SR). This protein has been shown to assemble both at the PM and at intracellular membranes of MT-4 cells and is competent for particle release (*Reil et al., 1998*). As can be seen in *Figure 1C*, PM localization and assembly site formation of Gag(Δ8–126 SR) was not affected by PI(4,5)P$_2$ depletion. Quantification of Gag clusters in DMSO (n = 15) or rCD1 treated (n = 18) cells showed no statistically significant difference in the number of Gag clusters at the ventral PM upon PI(4,5)P$_2$ depletion (p=0.8392) (*Figure 1F*). These results provide direct evidence for the requirement of the globular MA domain for PI(4,5)P$_2$ dependent Gag recruitment to the PM.

Next, we analyzed the relevance of PM PI(4,5)P$_2$ depletion for PM assembly site formation of two HIV-1 variants with mutations in the highly basic region of MA. These variants were previously characterized with respect to membrane interaction by in vitro liposome binding and membrane flotation assays. Changing MA residues K25,K26 to T25,T26 was reported to exhibit enhanced PI(4,5)P$_2$

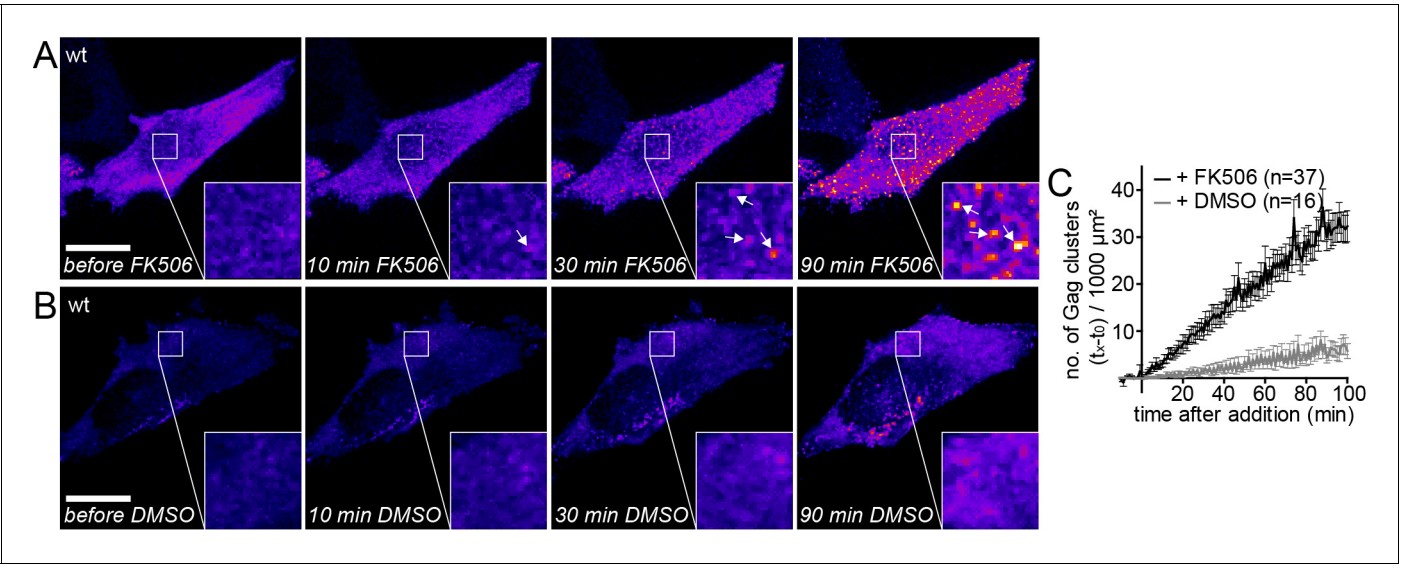

**Figure 2.** Gag assembly can be induced by PI(4,5)P$_2$ reconstitution. (A, B) Representative time-lapse SDC fluorescence images of the ventral PM of HeLa Kyoto cells transfected with plasmids expressing the rCDS and HIV-1 derived constructs pCHIV and pCHIV[EGFP]. Cells were treated with 1 µM rCD1 at 4 hpt and PI(4,5)P$_2$ rescue was induced at 22 hpt by addition of 1 µM FK506 (A) or 1% DMSO solvent control was added (B). Arrows indicate Gag-EGFP clusters. Scale bar represents 20 µm. (See also *Videos 2* and *3*) (C) Quantitative analysis of the increase in number of Gag-EGFP clusters at the PM following DMSO (grey) or FK506 (black) treatment. Error bars represent the standard error of the mean of n = 37 FK506 treated cells from four independent experiments and n = 16 DMSO treated cells from two independent experiments.

The following figure supplement is available for figure 2:

**Figure supplement 1.** The MA deletion mutant Δ8–126 SR does not respond to PI(4,5)P$_2$ depletion.

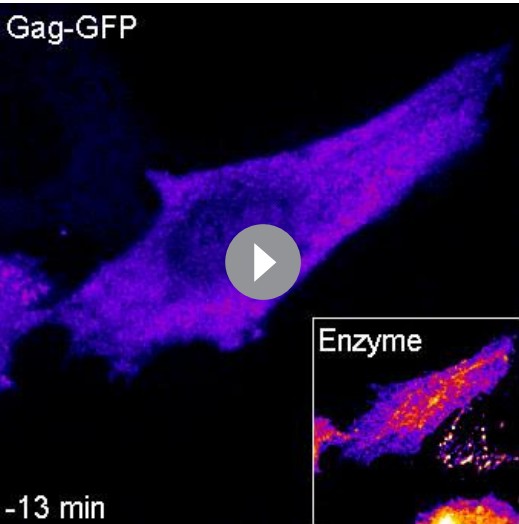

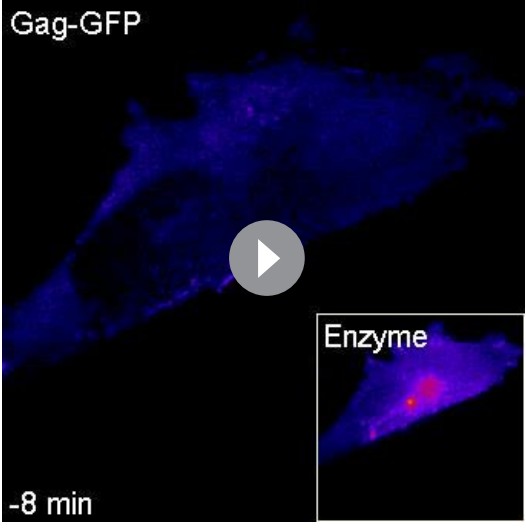

**Video 2.** Induction of HIV-1 assembly in PI(4,5)P$_2$ depleted cells by PI(4,5)P$_2$ reconstitution. Representative time-lapse SDC fluorescence microscopy sequence of the ventral PM of a HeLa Kyoto cell transfected with the rCDS and HIV-1 derived constructs pCHIV and pCHIV$^{EGFP}$. Cells were treated with 1 µM rCD1 at 4 hpt, followed by addition of 1 µM FK506 at 22 hpt. The sequence was acquired at a time resolution of 1 min/frame and is displayed with 12 fps. See *Figure 2A* for corresponding still images.

**Video 3.** Addition of DMSO solvent control to PI(4,5)P$_2$ depleted cells does not induce Gag assembly. Representative time-lapse SDC fluorescence sequence of the ventral membrane of a HeLa Kyoto cell transfected with the rCDS and HIV-1 derived constructs pCHIV and pCHIV$^{EGFP}$. Cells were treated with 1 µM rCD1 4 hpt, followed by addition of 1% DMSO solvent control at 22 hpt. The sequence was acquired at a time resolution of 2 min/frame and is displayed with 4.6 fps. See also *Figure 2B* for corresponding video still images.

independent membrane binding (*Chukkapalli et al., 2010*). Consistent with this in vitro observation, PM recruitment and assembly site formation of the 25/26KT variant was not significantly affected by PI(4,5)P$_2$ depletion in live cells (p=0.1242) (*Figure 1D and F*). Changing MA residues K29,K31 to E29,E31 had been reported to cause complete loss of PI(4,5)P$_2$ sensitivity with Gag mislocalization and impaired virus release; virus production was partially restored upon an additional change of MA residue E16 to K16 (*Tedbury et al., 2015*). This correlated with enhanced membrane binding of the MA variant in vitro, but the additional mutation did not appear to restore PI(4,5)P$_2$ dependence (*Tedbury et al., 2015*). In agreement with these results, we observed that PI(4,5)P$_2$ depletion did not have a statistically significant effect on assembly of 16EK 29/31KE Gag (p=0.5948) (*Figure 1E and F*).

## Synchronized induction of Gag assembly site formation upon PI(4,5)P$_2$ reconstitution

We then asked whether HIV-1 assembly site formation from cytosolic Gag molecules can be induced at the PM by restoring PI(4,5)P$_2$ levels. Similar to the previous experiments, PI(4,5)P$_2$ was depleted from the PM in rCDS expressing HeLa cells by rCD1 addition starting 4 hr after transfection. At 22 hr after transfection, Gag-EGFP was found to be mainly cytosolic with very few Gag assemblies at the PM (*Figure 2A*, left panel). Addition of FK506 and consequent restoration of PM PI(4,5)P$_2$ levels rapidly induced Gag cluster formation at the PM (*Figure 2A and C* and *Video 2*), whereas no major changes were observed upon addition of DMSO (*Figure 2B and C*, *Video 3*). After 90 min of FK506 treatment, the ventral PMs were covered with nascent HIV assembly sites and/or particles trapped between the cell and the coverslip (*Figure 2A*).

Analogous experiments performed with the MA deletion variant Δ8–126 SR confirmed dependence of the PI(4,5)P$_2$ responsive phenotype on the MA domain of Gag. Assembly site formation at the PM was not impaired for this variant in PM PI(4,5)P$_2$ depleted cells, and no significant difference

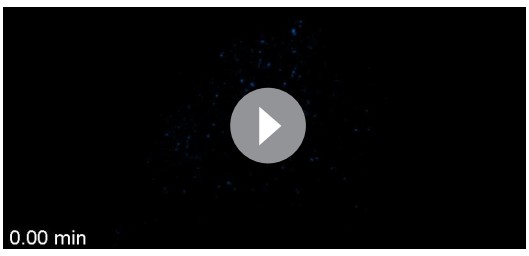

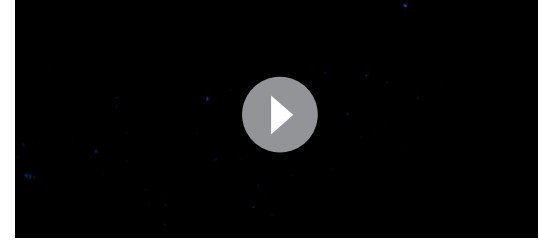

**Video 4.** High time resolution TIRF imaging of native Gag assembly. Representative time-lapse TIRF microscopy sequence of the ventral PM of a HeLa Kyoto cell transfected with the HIV-1 derived constructs pcHIV and pCHIV$^{EGFP}$. Cells were imaged at 22 hpt. The cell shows only few assembly sites in the beginning of the movie and progresses in assembly throughout the sequence. The sequence was acquired at a time resolution of 5 s/frame and is displayed with 140 fps. See *Figure 3*, showing the assembly kinetics derived from this and additional movies.

**Video 5.** High time resolution TIRF imaging of induced Gag assembly. Representative time-lapse TIRF microscopy sequence of the ventral PM of a HeLa Kyoto cell transfected with the rCDS and HIV-1 derived constructs pcHIV and pCHIV$^{EGFP}$. Cells were treated at 4 hpt with 1 µM rCD1 for 90 min, followed by addition of 1 µM FK506 at 22 hpt. The sequence was acquired at a time resolution of 5 s/frame and is displayed with 140 fps. See *Figure 3*, showing the assembly kinetics derived from this and additional movies.

was observed between cells subsequently treated with FK506 or DMSO (*Figure 2—figure supplement 1*). Deletion of the globular MA domain thus makes Gag independent of and unresponsive to PI(4,5)P$_2$.

In order to determine whether the rate of Gag assembly differed between induced and native Gag assembly sites, we applied Total Internal Reflection Fluorescence (TIRF) microscopy, which had been used to track single nascent HIV assembly sites over time with low background and high time resolution (*Jouvenet et al., 2008*; *Ivanchenko et al., 2009*). Cells were co-transfected with the rCDS components and pCHIV derivatives and either left untreated (*Video 4*), or subjected to PI(4,5)P$_2$ depletion at 4 hr after transfection followed by rescue of PI(4,5)P$_2$ PM levels by FK506 addition at 22 hr after transfection (*Video 5*). Fluorescence intensity was recorded over time for individual Gag assembly sites. *Figure 3A and B* show normalized and averaged HIV-1 assembly traces for 175 native and 186 induced assembly sites from three different cells each. No obvious difference in assembly behavior was apparent from these data sets. For a quantitative comparison, data from each individual trace were fitted to a single exponential equation to extract the assembly rate constant k (*Figure 3C*). Mean assembly rate constants derived from averaging all individual k values did not differ significantly between induced and native assembly, with k = 4.5 ± 0.47*$10^{-3}$ s$^{-1}$ for native and k = 3.49 ± 0.26*$10^{-3}$ s$^{-1}$ for induced assembly sites (p=0.8563) and a similar distribution of rate constants (*Figure 3—figure supplement 1A*). This translates into half-times of approximately 150 s and 200 s for native and induced assembly, respectively, or 8.5 min and 11 min for 90% completion of native and induced assembly, respectively. These values are in agreement with previously published data (*Ivanchenko et al., 2009*).

Visual inspection of the movies suggested that PM accumulation of assembly sites occurred more rapidly for induced compared to native assembly sites, while assembly rates at individual sites did not significantly differ between the two conditions. We therefore determined the number of native and induced PM assembly sites over time. *Figure 3D* shows that assembly sites accumulated faster and more synchronously when induced by PI(4,5)P$_2$ restoration compared to the native situation. Formation of 100 new assembly sites per 1000 µm$^2$ PM required only ~15 min for induced versus 45 min for native assembly.

Live cell TIRF imaging yielded assembly rates, but did not provide information on the morphology of induced assemblies, since the size of HIV-1 particles is below the diffraction limit of conventional light microscopy. Previous studies using PALM/dSTORM super resolution microscopy had revealed a distinct HIV-1 assembly site architecture, with compact Gag clusters of 100–120 nm diameter surrounded by larger patches of Env glycoprotein molecules recruited to the budding site in a Gag dependent manner (*Lehmann et al., 2011*; *Muranyi et al., 2013*). In order to obtain insight into the morphology of induced assembly sites, we applied Stimulated Emission Depletion (STED) super

resolution microscopy to analyze the distribution of Gag and Env at the ventral membrane of virus expressing cells with or without expression and induction of the rCDS. To allow for detection by STED nanoscopy, we employed Gag tagged with the stainable protein tag CLIP (*Gautier et al., 2008*; *Hanne et al., 2016*) instead of the EGFP tag, while Env was detected by indirect immunofluorescence. Imaging at the ventral PM does not distinguish late virus budding structures from trapped extracellular particles, and we therefore refer to late HIV-1 assemblies here.

As can be seen in *Figure 4*, STED nanoscopy strongly increased resolution compared to confocal images and HIV-1 assembly subdomains became discernible (compare confocal and STED panels). STED images revealed that neither the overall appearance of the cell nor the spatial distribution of Gag and Env at late assemblies differed notably between native and induced assembly. Both structures were characterized by a densely-packed Gag cluster with a diameter of about 120 nm, surrounded by a clustered accumulation of Env (*Figure 4*), in line with previous observations (*Muranyi et al., 2013*; *Roy et al., 2013*). The morphology of induced late assemblies was indistinguishable from native sites in micrographs (compare *Figure 4A and B*) as well as in averaged line profiles of individual assemblies (compare *Figure 4C and D*).

While the previous experiments established that induced Gag assemblies appeared indistinguishable from their native counterparts, the recovery of infectious virus would be ultimate proof for synchronized induction of functional HIV-1 assembly. To this end, we established a stable HeLa$_{rCDS}$ cell line constitutively expressing Anchor and Enzyme tagged with autofluorescent proteins. The presence of both components of the rCDS in all cells prior to transfection with the HIV-1 expressing plasmid prevents constitutive production of infectious virus from cells only transfected with the HIV-1 plasmid, but lacking one or both components of the rCDS, while an uneven distribution of components between cells is expected to occur when the three plasmids are cotransfected. As shown in *Figure 5A*, HeLa$_{rCDS}$ cells constitutively express both fusion proteins, with the Enzyme detected in the cytoplasm and the Anchor at the PM. Next, we assessed functionality of the rCDS in these cells by expressing EGFP-PLCδ-PH, whose PM localization serves as indicator for PI(4,5)P$_2$. In the absence of rCD1, the indicator was detected mainly at the PM (*Figure 5B*, left panel). Upon rCD1 addition (*Figure 5B*, middle panel), the Enzyme translocated to the PM (see insert), resulting in relocalization of the indicator protein to the cytoplasm as a consequence of PI(4,5)P$_2$ depletion from the PM. This effect was reversed upon addition of FK506 (*Figure 5B*, right panel).

After validation of the stable cell line, constitutive or induced virus release and formation of infectious HIV-1 was analyzed following transfection of HeLa$_{rCDS}$ cells with the complete proviral plasmid pNL4-3. rCD1 was added 4 hr after transfection. Comparing virus production from rCD1 and control cells at 22 hr after transfection showed a strong inhibition by PI(4,5)P$_2$ depletion (80–85% reduction compared to control cells) (*Figure 5C*; see also *Figure 5—figure supplement 1A* for non-normalized data). Virus assembly was then induced in rCD1-treated cells by addition of 1 μM FK506 at 22 hr after transfection. Extracellular virus was collected 2 hr after addition of FK506 since a short harvest time limits the analysis mainly to virions formed by induced assembly. Virus production was restored after FK506 addition (*Figure 5—figure supplement 1A*). Most importantly, relative infectivity of virions produced under inducing conditions was only mildly reduced (~65% compared to virions produced under native conditions from DMSO treated HeLa$_{rCDS}$ cells) (*Figure 5D*; see also *Figure 5—figure supplement 1B and C* for non-normalized data). These results indicate that induced PM recruitment of Gag allows faithful assembly of infectious HIV-1 particles.

## PI(4,5)P$_2$ is required to retain the partially assembled Gag lattice at the PM

Our results so far confirmed the essential role of PI(4,5)P$_2$ for Gag recruitment to the PM and established an inducible system for assembly of infectious HIV-1. To determine whether PI(4,5)P$_2$ is also required to retain multimeric Gag assemblies at the PM, we performed reversible depletion of PM PI(4,5)P$_2$ in cells displaying pre-formed HIV-1 assembly sites. Assuming insertion of the N-terminally attached myristic acid into the inner membrane leaflet for most or all Gag molecules in nascent assembly sites, PI(4,5)P$_2$ would not be expected to play a role at this stage. Rapid and reversible PM PI(4,5)P$_2$ depletion in living cells offers a unique opportunity to address this issue.

HIV-1 expressing HeLa cells carrying the rCDS were incubated without compound addition until a large number of Gag assemblies had accumulated at the PM. Gag assemblies at the ventral PM (towards the coverslip) comprise nascent assembly sites as well as extracellular particles trapped

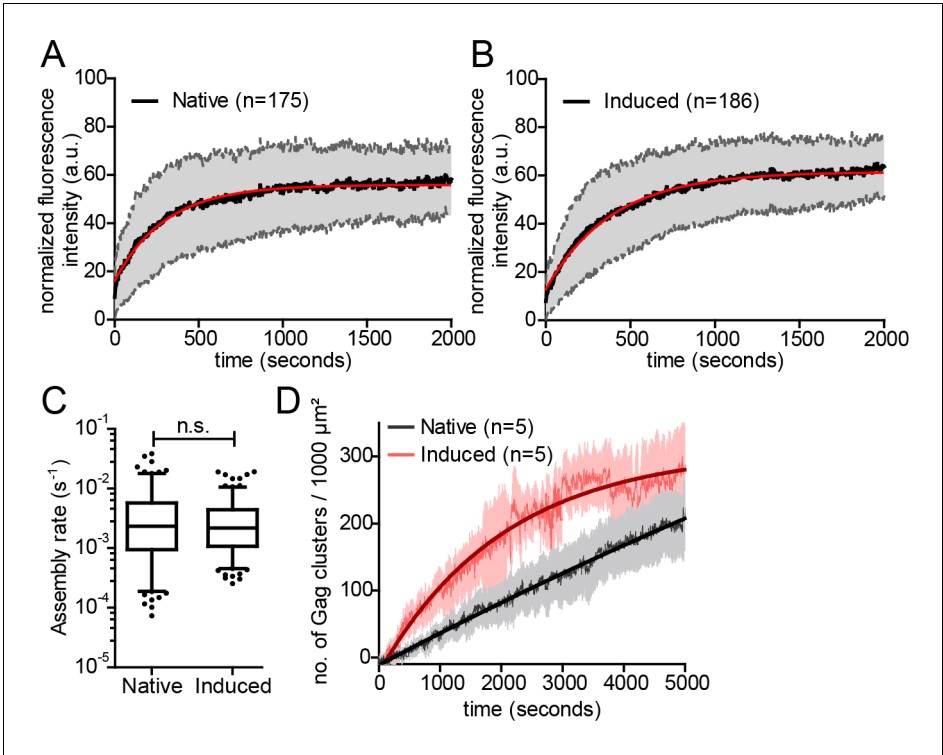

**Figure 3.** Kinetics of native and induced Gag assembly do not differ significantly. (**A, B**) Normalized and averaged HIV-1 assembly traces of 175 individual native (**A**) and 186 individual induced (**B**) assembly sites at the ventral PM of n = 3 cells each. The standard deviation is shown in grey, while a single exponential fit is shown in red. (**C**) Assembly rate constants derived from the individual assembly traces by fitting to single exponential equations. Mean assembly rate constants for native and induced assembly were $4.5 \pm 0.47*10^{-3}$ s$^{-1}$ and $3.49 \pm 0.26*10^{-3}$ s$^{-1}$, respectively. Whiskers plots represent 5–95 percentile (statistical significance was assessed with the Mann-Whitney U test; differences were considered significant when p≤0.05). (**D**) Quantitative analysis of nascent native and induced Gag clusters (imaged by high time-resolution TIRF microscopy) forming at the ventral membrane over time. Numbers represent the mean amount of Gag-GFP clusters per 1000 µm$^2$ membrane. Error bars represent SEM. (See also **Figure 3—figure supplement 1** for relative frequency distributions of assembly rates and **Videos 4** and **5**).

The following figure supplement is available for figure 3:

**Figure supplement 1.** Relative frequency distributions of assembly rates do not differ between native and induced assembly.

between the cell and the substrate. These extracellular particles can obviously not be affected by rCD1 mediated PI(4,5)P$_2$ PM depletion, and microscopy of the ventral PM is therefore not suitable for such an experiment. Instead, a central cell section was imaged in the subsequent experiments to focus on nascent assembly sites still connected to the cytosol.

PI(4,5)P$_2$ depletion was induced in cells with numerous Gag assembly sites by addition of rCD1. Unexpectedly, the vast majority of cells (52 of 59) lost most pre-formed Gag assemblies from the lateral PM upon PI(4,5)P$_2$ depletion (**Figure 6A** and **Video 6**). For quantitative analysis, we determined the relative number of Gag clusters over time following PI(4,5)P$_2$ depletion in the cell shown in **Figure 6A** (**Figure 6C**). This number decreased to 50% at ~30 min after rCD1 addition and reached a minimum level of ~30%. Averaging the relative amount of Gag clusters detected over time in n = 10 cells yielded a very similar result (**Figure 6E**). In contrast, DMSO treated cells retained or slowly increased the number of Gag assembly sites at the lateral PM over time (**Figure 6C and E**, **Figure 6—figure supplement 1A**, **Video 7**), excluding any unspecific effect of the carrier or the imaging procedure. PM PI(4,5)P$_2$ depletion also had no effect on preformed assembly sites for the

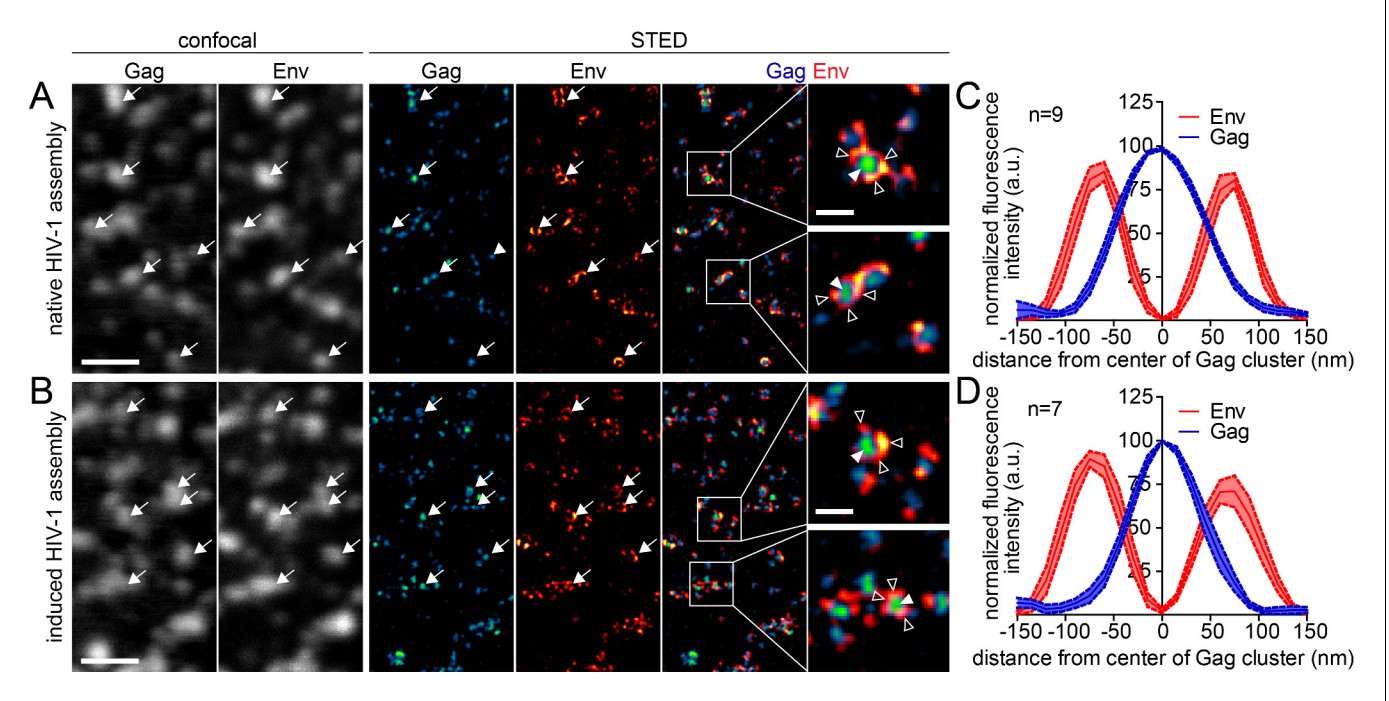

**Figure 4.** Native and induced HIV-1 assembly sites are indistinguishable by STED nanoscopy. (A, B) Confocal (left) and STED images (right) of the ventral PM of HeLa Kyoto cells transfected only with the HIV-1 derived constructs pCHIV and pCHIV^CLIP (native assembly, (**A**)) or additionally with the plasmids expressing the rCDS (induced assembly, (**B**)). In B, HIV-1 assembly was first inhibited by addition of rCD1 at 4 hpt and then induced by addition of FK506 for 90 min at 22 hpt. Gag (cyan) was detected via Atto 590 CLIP and Env (red) was detected via indirect immunolabeling. Some individual assembly sites are highlighted with arrows. Gag clusters (filled arrowheads) were surrounded by Env accumulations (open arrowheads). Scale bar represents 1 µm (overview images) or 200 nm (enlargements). (**C, D**) Averaged line profiles of selected native (**C**) or induced (**D**) Gag assembly sites (n = 9 and n = 7, respectively). Error bars represent SEM.

$\Delta 8$–126 SR Gag variant (*Figure 6—figure supplement 1B and C*), and this was confirmed by quantitative analysis of Gag($\Delta 8$–126 SR) cluster number over time (*Figure 6—figure supplement 1D*).

The number of Gag assembly sites at the plasma membrane at a given time reflects the equilibrium between assembly site formation and loss of assembly sites, either by extracellular release of budded virions or by dissociation of pre-formed assembly structures from the PM into the cytosol. To assess the contribution of virus release to the observed reduction of PM assemblies following PI(4,5)P$_2$ depletion, we performed the depletion experiment with a Vpu-defective HIV-1 derivative. Vpu counteracts the host restriction factor tetherin (*Figure 6—figure supplement 2A and B*). In the absence of Vpu, tetherin retains fully budded HIV-1 particles at the PM and thereby inhibits virus release (*Neil et al., 2008*; *Van Damme et al., 2008*). High amounts of PM tetherin in cells producing Vpu(-) particles should therefore largely prevent virus release, while removal of assembly sites from the PM to the cytosol would be unaffected. Control experiments confirmed that neither the presence of the rCDS nor its activation affected the localization of tetherin (*Figure 6—figure supplement 2C and D*). Importantly, the loss of Gag clusters from the PM upon PI(4,5)P$_2$ depletion was unaffected by lack of Vpu (*Figure 6—figure supplement 2E and F*), supporting the conclusion that Gag from nascent assembly sites re-localizes to the cytosol upon PI(4,5)P$_2$ depletion.

The observed reduction of Gag clusters at the PM upon PI(4,5)P$_2$ depletion could also be caused by constitutive loss of Gag from the PM, which is normally overcome by PM binding of newly synthesized Gag molecules. In this case, one would expect a reduction of preformed assembly sites when blocking synthesis of new proteins by addition of cycloheximide (CHX). Adding CHX to cells with established Gag clusters did not affect the appearance or number of Gag assemblies over up to 90 min (*Figure 6—figure supplement 2G and H*), suggesting that loss of Gag assemblies from the cell membrane is caused by PI(4,5)P$_2$ dependence of their PM association.

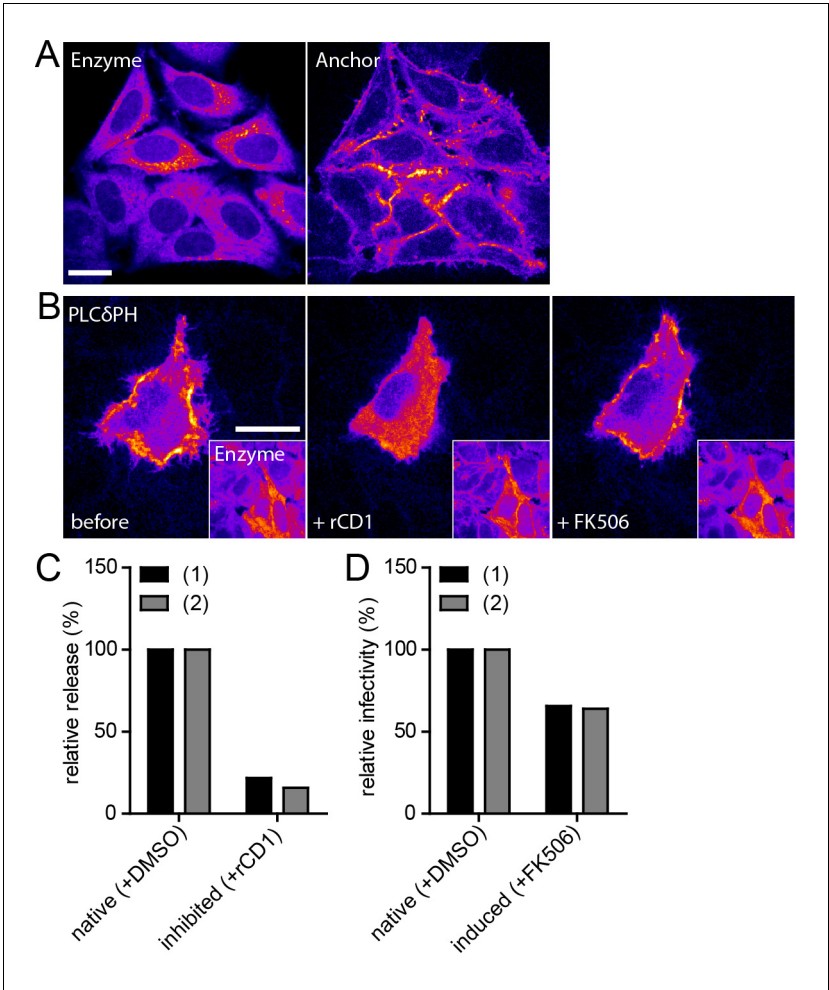

**Figure 5.** Infectivity of virions produced under native and induced conditions in HeLa_{rCDS} cells. (**A**) Representative SDC fluorescence images of the central section of HeLa_{rCDS} cells expressing the Anchor and Enzyme. Single planes are shown. (**B**) HeLa_{rCDS} cells were transfected with pEGFP-PLCδ-PH and imaged 22 hpt (left panel). Cells were treated with 1 μM rCD1 (middle panel), followed by addition of 1 μM FK506 (right panel). Maximum intensity projections of three focal planes acquired with an axial spacing of 0.5 μm are shown. Scale bars in A and B represent 20 μm. (**C,D**) HeLa_{rCDS} cells were transfected with pNL4-3 and treated with DMSO (native) or 1 μM rCD1 (inhibited) 5 hpt. (**C**) Relative RT activity as a measure of virus release was determined from supernatants harvested at 22 hpt. (**D**) Cells were subsequently treated with DMSO (native) or 1 μM FK506 (induced). Infectivity on TZM-bl reporter cells was determined from supernatants harvested 2 hr after addition of DMSO (native)/1 μM FK506 (induced) and used to calculate normalized relative infectivity values. Data from two independent experiments [experiments (1) and (2)] are shown. Please refer to *Figure 5—figure supplement 1* for non-normalized data.

The following figure supplement is available for figure 5:

**Figure supplement 1.** Release, infectivity and relative infectivity of virions produced under native and induced conditions.

## HIV-1 Gag assembly sites can be reconstituted by rescue of PM PI(4,5)P_2

We next asked whether Gag molecules that had been dissociated from PM assembly sites by PI(4,5)P_2 depletion could form new assembly sites upon restoration of PM PI(4,5)P_2. For this experiment, we treated cells displaying pre-formed assembly sites with rCD1 for 90 min as described above, followed by addition of FK506 to allow PM PI(4,5)P_2 reconstitution. FK506 treatment yielded a rapid increase in the number of Gag assembly sites at the PM (*Figure 6B*, *Video 6*). The number of Gag

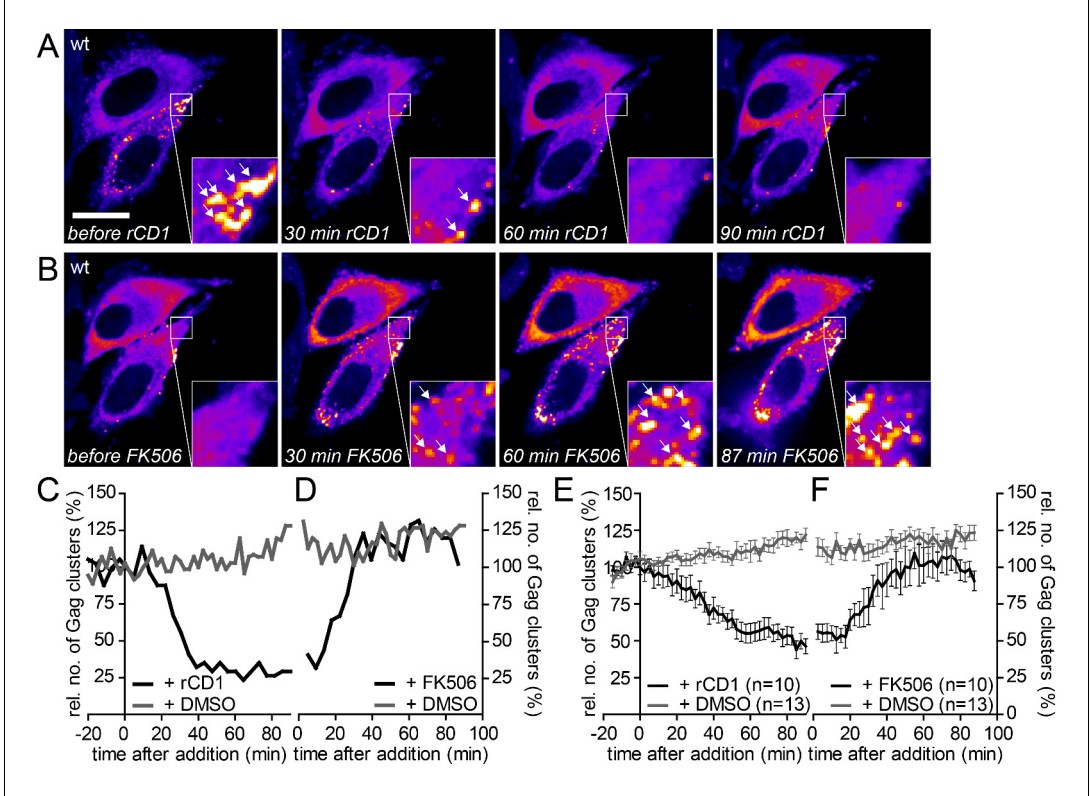

**Figure 6.** The partially assembled Gag lattice dissociates reversibly upon PI(4,5)P$_2$ depletion. (**A, B**) Representative time-lapse SDC fluorescence images of the central volume of HeLa Kyoto cells transfected with plasmids expressing the rCDS and HIV-1 derived constructs pCHIV and pCHIV$^{EGFP}$. Maximum intensity projections of four focal planes acquired with an axial spacing of 0.5 μm are shown. Cells were treated at 22 hpt with 1 μM rCD1 for 90 min (**A**) and FK506 was added subsequently (**B**). Arrows indicate Gag-EGFP clusters. Scale bar represents 20 μm. (**C, D**) Relative number of Gag clusters in the cell shown in (**A, B**) and the control cell shown in *Figure 6—figure supplement 1A* following rCD1 (black) or DMSO (grey) (**C**) and FK506 (black) or DMSO (grey) (**D**) addition, respectively. (**E, F**) Quantitative analysis of the mean relative number of Gag-EGFP clusters following rCD1 (black) or DMSO (grey) (**E**) and FK506 (black) or DMSO (grey) (**F**) addition, respectively. Error bars represent SEM for n = 10 rCD1+FK506 treated cells from six independent experiments and n = 13 DMSO treated cells from four independent experiments. (See also *Videos 6* and *7*).

The following figure supplements are available for figure 6:

**Figure supplement 1.** Addition of DMSO solvent control does not affect already assembled Gag clusters and the MA deletion mutant *Δ8–126 SR* does not respond to PI(4,5)P$_2$ depletion and rescue.

**Figure supplement 2.** Influence of rate of particle release and new particle formation on number of Gag clusters.

**Figure supplement 3.** G2AGag does not respond to PI(4,5)P$_2$ rescue following PI(4,5)P$_2$ depletion.

**Figure supplement 4.** Kinetics of reinduced and native assembly at the lateral PM.

clusters reached the initial level (prior to rCD1 mediated PI(4,5)P$_2$ depletion) at 30–40 min after addition of FK506 (*Figure 6D and F*). No effect on the number of Gag clusters was observed in control cells treated with DMSO (*Figure 6—figure supplement 1A*, *Figure 6D and F*, *Video 7*) or when the PI(4,5)P$_2$ independent MA variant *Δ8–126 SR* was used (*Figure 6—figure supplement 1B, C and E*). Visual inspection of movie sequences indicated that newly induced assembly sites generally did not appear at those positions where previously dissociated sites had been observed.

Quantitative analyses indicated that accumulation of reinduced HIV-1 Gag assemblies following a cycle of PI(4,5)P$_2$ depletion and restoration (*Figure 6D and F*) occurred twice as fast as initial accumulation of budding sites under induced conditions (compare *Figure 2C*). To analyze whether reinduced Gag assembly sites at the PM upon PI(4,5)P$_2$ reconstitution comprised only newly synthesized

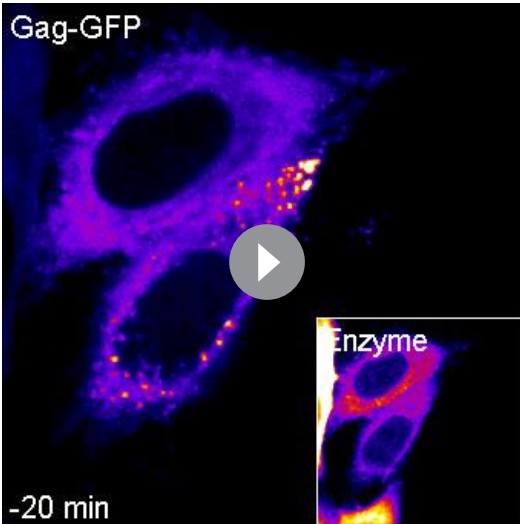

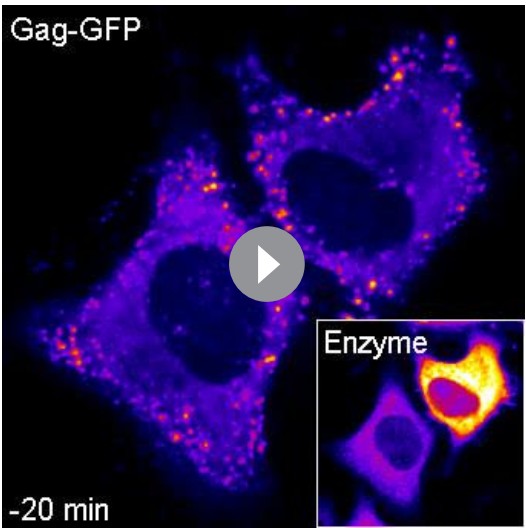

**Video 6.** The partially assembled Gag lattice dissociates reversibly upon PI(4,5)P$_2$ depletion. Representative time-lapse SDC fluorescence microscopy sequence of the central volume of a HeLa Kyoto cell transfected with the rCDS and HIV-1 derived constructs pcHIV and pCHIV$^{EGFP}$. Maximum intensity projections of four focal planes acquired with an axial spacing of 0.5 µm are shown. Cells were treated at 22 hpt with 1 µM rCD1 for 90 min, followed by addition of 1 µM FK506. The sequence was acquired at a time resolution of 4 min/frame and is displayed with 4 fps. See ***Figure 6A and B*** for corresponding still images. DOI: 10.7554/eLife.25287.022

**Video 7.** Addition of DMSO solvent control does not affect already assembled Gag clusters. Representative time-lapse SDC fluorescence microscopy sequence of the central volume of HeLa Kyoto cells transfected with the plasmids expressing the rCDS and HIV-1 derived constructs pcHIV and pCHIV$^{EGFP}$. Maximum intensity projections of four focal planes acquired with an axial spacing of 0.5 µm are shown. Cells were treated at 22 hpt with 1% DMSO for 90 min, followed by addition of additional 1% DMSO. The sequence was acquired at a time resolution of 2.5 min/frame and is displayed with 8 fps. See ***Figure 6—figure supplement 1A*** for corresponding still images. DOI: 10.7554/eLife.25287.023

Gag molecules or whether 'recycled' molecules previously depleted from the membrane contribute, we blocked protein translation by cycloheximide (CHX) treatment 60 min prior to rCD1 addition. As shown in ***Figure 7A and C***, PI(4,5)P$_2$ depletion resulted in loss of clusters from the PM that was fully reverted by restoring PI(4,5)P$_2$ levels in the presence of CHX (***Figure 7B and D***). These data indicate that Gag molecules synthesized before assembly site dissociation are sufficient for formation of new assembly sites following PI(4,5)P$_2$ reconstitution and new protein synthesis is not required.

Based on these observations, we asked whether the more rapid accumulation of assembly sites upon dissociation and reinduction may be explained by multimeric Gag lattice remnants following assembly site dissociation. To address this issue, we employed a Gag variant lacking the N-terminal myristoylation site (Gag G2A). This variant cannot assemble at the PM (***Bryant and Ratner, 1990***; ***Göttlinger et al., 1989***), but can be rescued by a low amount of co-expressed wild-type Gag (***Park and Morrow, 1992***; ***Morikawa et al., 1996***). Accordingly, co-expression of Gag-EGFP(G2A) and wild-type Gag (in 2:1 molar ratio) led to formation of fluorescent assembly sites at the PM (***Figure 6—figure supplement 3A***) of cells expressing the Anchor and Enzyme, while Gag-EGFP(G2A) alone failed to assemble. As expected, addition of rCD1 resulted in a loss of Gag-EGFP(G2A) clusters from the PM (***Figure 6—figure supplement 3A and C***). However, only few Gag-EGFP(G2A) clusters reformed at the PM and the quantitative analysis revealed, that Gag-EGFP(G2A) was not efficiently re-recruited to the PM upon PI(4,5)P$_2$ rescue (***Figure 6—figure supplement 3B and D***), while wild-type Gag readily formed new assembly sites under these conditions (***Figure 6***). This result is consistent with complete dissociation of Gag clusters following PI(4,5)P$_2$ depletion. Quantitative analysis revealed that neither the size nor the fluorescence intensity of residual clusters increased following PI(4,5)P$_2$ restoration (***Figure 6—figure supplement 3E–H***), arguing against recruitment of Gag(G2A) molecules to residual assembly sites.

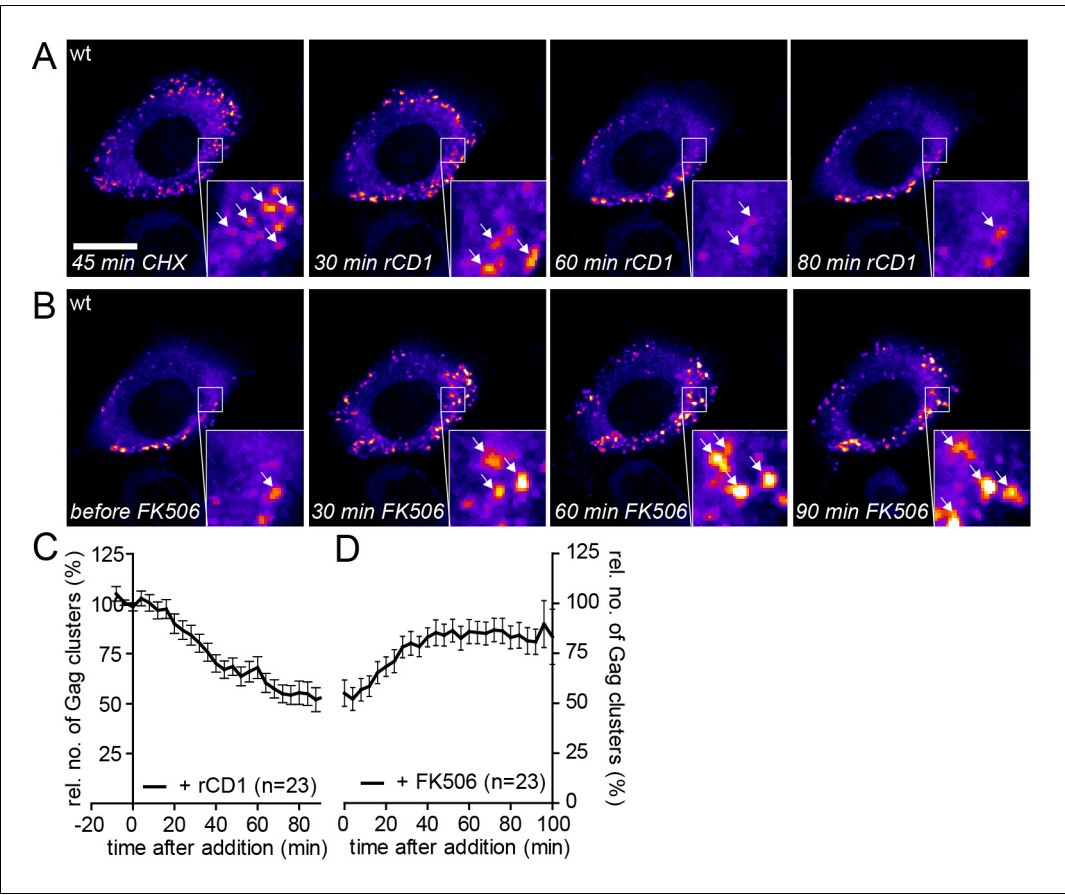

**Figure 7.** Assembly of Gag can be reinduced in the presence of cycloheximide. (**A**, **B**) Representative time-lapse SDC fluorescence images of the central volume of a HeLa Kyoto cell transfected with plasmids expressing the rCDS and HIV-1 constructs pCHIV and pCHIV^EGFP. Maximum intensity projections of four focal planes acquired with an axial spacing of 0.5 µm are shown. Cells were pre-treated at 21 hpt with 10 µg/ml CHX for 60 min. Subsequently cells were treated with 1 µM rCD1 for 90 min (**A**), followed by addition of FK506 (**B**). Arrows indicate individual Gag-EGFP clusters. Scale bar represents 20 µm. (**C**, **D**) Relative number of Gag-EGFP clusters over time following rCD1 (**C**) or FK506 (**D**) treatment. Error bars represent SEM of data from n = 23 cells from three independent experiments.

To compare the dynamics of assembly site formation for induced and reinduced assembly, we analyzed assembly rates following $PI(4,5)P_2$ depletion and reconstitution. Spinning disk confocal microscopy was performed at high time resolution in a 3D volume to identify and track single assembly sites at the lateral PM (*Video 8*). *Figure 6—figure supplement 4A* shows normalized and averaged traces of 96 individual reinduced assembly sites from three cells.

For a quantitative comparison, data from each individual trace were fitted to a single exponential equation to extract the assembly rate constant k (*Figure 6—figure supplement 4C*). The mean assembly rate constant of $k = 5.8 \pm 0.6 \times 10^{-3}$ $s^{-1}$ for reinduced assembly, which was derived from averaging all individual k values, was slightly, but significantly, increased compared to native assembly (native: $k = 4.5 \pm 0.5 \times 10^{-3}$ $s^{-1}$, p<0.0001) (*Figure 6—figure supplement 4C*). The assembly constant for reinduced assembly translates into a half-time of ~120 s (native: $t_{1/2}$ ~150 s). Native and induced assembly kinetics were both determined at the ventral membrane, while reinduced assembly had to be measured at the lateral membrane, and this difference could influence the observed assembly rates. We therefore determined assembly kinetics of native Gag assembly site formation at the lateral membrane as well. *Figure 6—figure supplement 4B* shows normalized and averaged traces of 169 individual native assembly sites at the lateral PM of six cells. The average assembly constant derived from single exponential fits for all native assembly traces under these conditions of

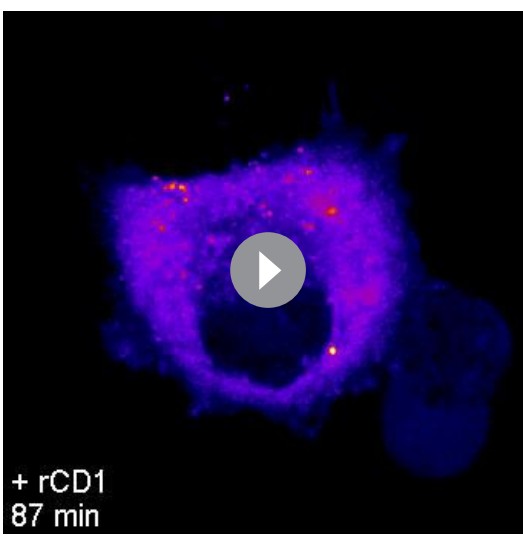

**Video 8.** High time resolution SDC imaging of reinduced Gag assembly. Representative time-lapse SDC fluorescence microscopy sequence of the central volume of a HeLa Kyoto cell transfected with the plasmids expressing the rCDS and HIV-1 derived constructs pcHIV and pCHIV[EGFP]. Maximum intensity projections of four focal planes acquired with an axial spacing of 0.5 µm are shown. Cells were treated at 22 hpt with 1 µM rCD1 for 90 min, followed by addition of 1 µM FK506. The sequence was acquired at a time resolution of 6 s/frame and is displayed with 100 fps. See *Figure 6—figure supplement 4*, showing the assembly kinetics derived from this and additional movies.

$k = 5.9 \pm 0.4 * 10^{-3}$ s$^{-1}$ (*Figure 6—figure supplement 4C*) was very similar to that observed for reinduced assembly (no significant change, p=0.1569) and was again significantly faster than observed for assembly at the ventral membrane (p<0.0001). The half-time of native assembly at the lateral PM was changed accordingly ($t_{1/2}$ ~115 s). The observed difference may reflect effects of the adjacent glass substrate on the kinetics of HIV-1 Gag assembly at the ventral membrane, but clearly shows that reinduced and native assembly site formation occur with very similar kinetics. The distribution of rate constants also did not reveal any major difference between reinduced and native assembly (*Figure 6—figure supplement 4D*).

Finally, we asked whether the reinduced HIV-1 Gag assemblies were morphologically normal using STED nanoscopy. Since depletion from the ventral PM was inefficient, we again focused on the lateral PM for these analyses. As shown in *Figure 8A*, late native HIV-1 assemblies at the lateral PM appeared very similar to those detected at the ventral PM (compare *Figure 4*). In both cases, condensed Gag clusters with a diameter of roughly 120 nm surrounded by larger Env clusters were detected in untreated cells. Reinduced assembly sites also displayed this distinct phenotype (*Figure 8B*) and were undistinguishable from native assembly sites at the lateral PM with respect to diameter and morphology in micrographs and line plot diagrams of Gag and Env (*Figure 8C and D*).

## Discussion

PM association of HIV-1 Gag - and consequently assembly site formation and virus production - have previously been shown to depend on PM PI(4,5)P$_2$. Here, we made use of a recently established chemical biology tool allowing rapid and reversible manipulation of PI(4,5)P$_2$ levels at the PM in living cells (*Feng et al., 2014*; *Schifferer et al., 2015*) to study the role of PI(4,5)P$_2$ for HIV-1 assembly site formation and maintenance. This rCDS has several advantages compared to alternative strategies. First, PI(4,5)P$_2$ depletion is strictly dependent on rCD1 addition and can be rapidly achieved within minutes. This allows maintaining normal cell metabolism prior to compound addition despite the presence of all rCDS components in the cell, and should thus prevent induction of escape pathways. Second, the system is completely and rapidly reversible, allowing restoration of PM PI(4,5)P$_2$ levels within minutes of FK506 addition. Here, we show that long-term depletion of PM PI(4,5)P$_2$ is compatible with cell viability and viral protein expression, while completely preventing Gag membrane targeting and assembly site formation. Furthermore, PI(4,5)P$_2$ depletion also caused loss of pre-assembled Gag lattices from the PM and large clusters removed from the membrane appeared to dissociate into monomers or small oligomers in the cytosol. Restoration of PM PI(4,5)P$_2$ in these cells reinduced assembly site formation even in the absence of new protein synthesis indicating that the dissociated Gag molecules remained assembly competent. These results reveal an important role of PI(4,5)P$_2$ beyond Gag recruitment to the PM and suggest that Gag interactions with PM lipids may be more dynamic than previously thought.

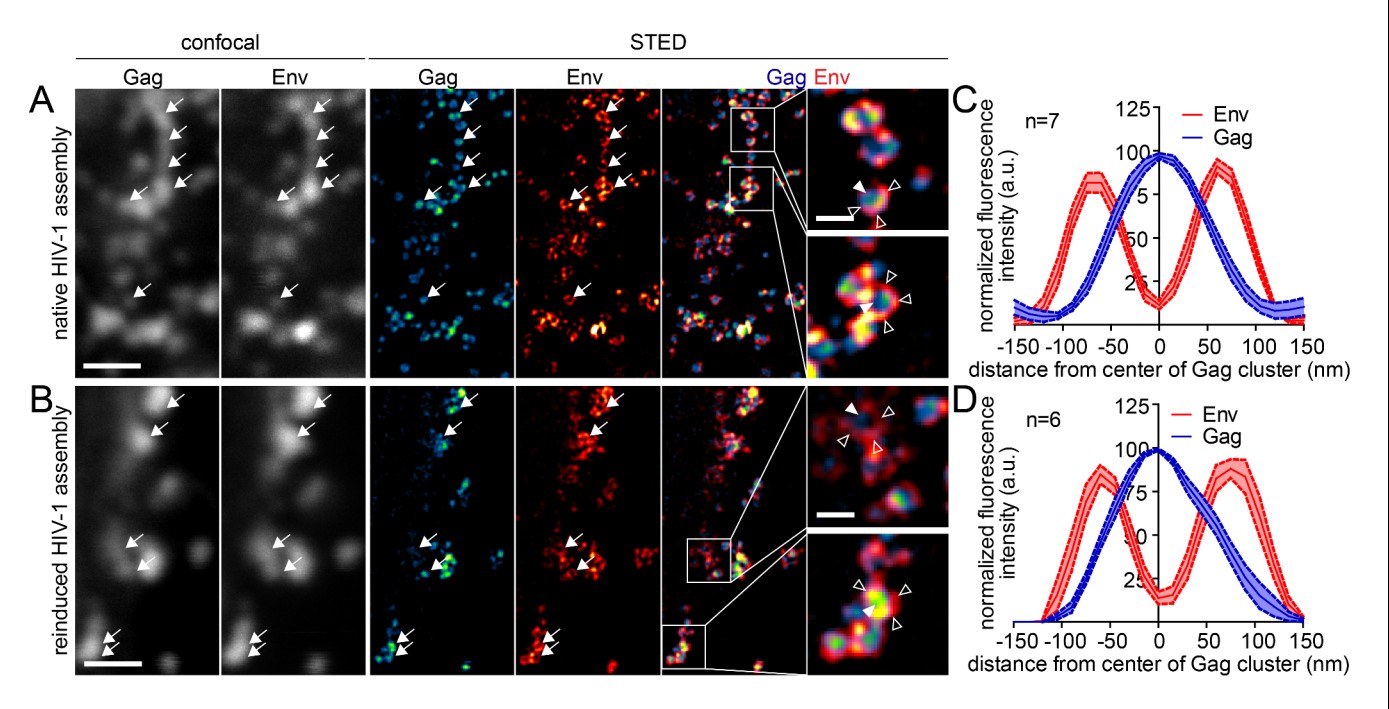

**Figure 8.** Native and reinduced HIV-1 assembly sites are indistinguishable by STED nanoscopy. (A, B) Confocal (left) and STED images (right) of the lateral PM of fixed HeLa Kyoto cells expressing only the HIV-1 constructs pCHIV and pCHIV^CLIP (native assembly, (A)) or in addition the plasmids expressing the rCDS (reinduced assembly, (B)). In B, dissociation of assembled Gag clusters was induced by addition of 1 μM rCD1 for 90 min at 22 hpt; subsequently Gag assembly was reinduced by addition of 1 μM FK506 for 90 min. Gag (cyan) was detected via Atto 590 CLIP and Env (red) was detected via indirect immunolabeling. Some individual HIV-1 assembly sites are highlighted by arrows. Gag clusters (filled arrowheads) were surrounded by Env accumulations (open arrowheads). Scale bar represents 1 μm (overview images) or 200 nm (enlargements). (C, D) Averaged line profiles of selected native (C) or induced (D) Gag assembly sites (n = 7 and n = 6 for native and reinduced assembly sites, respectively). Error bars represent SEM.

PI(4,5)P$_2$ and MA-dependent PM targeting of Gag and assembly site formation is consistent with previous studies showing that continuous overexpression of 5Ptase causes loss of Gag targeting in Gag-producing cells, that Gag-derived proteins preferentially interact with PI(4,5)P$_2$ in vitro in an MA-dependent manner and that PI(4,5)P$_2$ was found to be enriched in virions (*Ono et al., 2004*; *Saad et al., 2006*; *Chukkapalli et al., 2008*; *Chan et al., 2008*). Applying an inducible system in the current study allowed us to investigate the kinetics of assembly site formation. Our results show that rate constants for native and induced assembly, and even for reinduced assembly following depletion and subsequent restoration of PI(4,5)P$_2$, are virtually identical. Given that a large pool of myristoylated and assembly competent Gag is present in the cells prior to induction (or reinduction), one may have expected assembly to occur faster than under native conditions, but this was not the case. This observation is consistent with the assumption that PI(4,5)P$_2$ is important to retain Gag once it reaches the PM, but does not affect its PM trafficking, which may be rate-limiting under these conditions. Interestingly, Gag assembly site formation was not faster, but more synchronous under induced or reinduced assembly conditions compared to native assembly. This finding most likely reflects the presence of a large pool of assembly competent Gag in PI(4,5)P$_2$ depleted cells, which can only be retained at the PM upon PI(4,5)P$_2$ restoration. Gag molecules may transiently associate with all cellular membranes including the PM in the absence of PI(4,5)P$_2$, but are stably retained only in the presence of this phosphoinositide. Assembly site formation therefore is synchronized by the rCDS, which provides a promising tool to study the role of host components during HIV-1 assembly and the induction of virus maturation. Analysis of these processes is so far limited by the inherently asynchronous nature of virus production – between cells and within an individual cell – and synchronizing this process *via* the described approach can overcome this limitation. The observation that

virions released under those conditions are infectious and the opportunity to use stable HeLa$_{rCDS}$ cells for bulk assays broadens the range of applications for this system.

The finding that PI(4,5)P$_2$ is also required to retain partially or completely assembled Gag lattices at the PM was unexpected, since a triggered myristoyl switch in conjunction with ionic interactions between the HBR and negatively charged lipids of the inner leaflet would be expected to retain Gag at the PM, especially considering the high avidity of the multimeric Gag lattice (ca. 2,500 Gag molecules per assembly site). Large PM Gag clusters resembling fully assembled HIV-1 budding structures were rapidly and completely lost from the PM upon PI(4,5)P$_2$ depletion, however. Thus, PI(4,5)P$_2$ is clearly required throughout assembly and budding of HIV-1 particles and not only for Gag membrane targeting. In vitro, MA binding to liposomes is significantly weaker for PI(4)P compared to PI(4,5)P$_2$ (*Mercredi et al., 2016*), suggesting that Gag dissociation from the PM may be directly caused by dephosphorylation of PI(4,5)P$_2$ and consequent loss of Gag interaction. It is not entirely clear, however, whether Gag-associated phosphoinositides are good substrates for the phosphatase since the enzyme would have to enter the tight protein lattice of the assembling virion. Accordingly, effects of general depletion of PM PI(4,5)P$_2$ with lipid exchange of Gag molecules must also be considered.

Our results are not easily reconciled with the suggested lipid flip, where the 2′acyl chain of PI(4,5)P$_2$ would be pulled from the membrane and exchanged for the myristic acid once the myristate switch is triggered. As discussed by Summers and colleagues (*Mercredi et al., 2016*), extrusion of the 20-carbon 2′-acyl chain of PI(4,5)P$_2$ would require ~24 kcal/mol (*Tanford, 1979*; *Israelachvili et al., 1977*), while inserting 10 carbons of myristate into the lipid bilayer would gain only 8 kcal/mol. More importantly, our results appear inconsistent with the continuous insertion of the myristate moiety of all or the majority of Gag molecules into the inner leaflet. While the binding energy of the myristate anchor is not sufficient to retain individual Gag molecules at the PM, the avidity effects of up to 2500 Gag molecules with all or the majority of their N-terminal myristate groups inserted into the inner leaflet should make membrane association of the Gag lattice almost irreversible, independent of PI(4,5)P$_2$. Given that the binding energy of myristate interaction with MA is the same as for extrusion of myristate from the PM (8 kcal/mol) (*Charlier et al., 2014*), we suggest a dynamic equilibrium between fully membrane-inserted and MA-bound PI(4,5)P$_2$ molecules at the assembly site. Accordingly, myristate may readily flip between the inner leaflet of the PM and the lipid binding pocket of MA. Once Gag reaches the PM, PI(4,5)P$_2$ association triggers the myristate switch, and this effect is enhanced by the presence of acidic phospholipids. The combination of specific PI(4,5)P$_2$ binding and myristate insertion into the PM then anchors the Gag molecule and makes it available for virus assembly. Subsequently, a dynamic exchange of lipid interactions may occur, where depletion of PM PI(4,5)P$_2$ causes loss of Gag molecules once their myristate moieties flip out from the membrane.

Finally, our results also shed some light on the question whether Gag assembly is directed to pre-existing assembly-prone membrane microdomains or whether Gag creates its own microdomain. Comparison of the membrane localization of Gag assembly sites prior to PI(4,5)P$_2$ depletion with assembly sites in the same region of the cell after PI(4,5)P$_2$ restoration indicated that they occur in different positions. The conclusion that there are no stable assembly-prone membrane microdomains is consistent with earlier observations showing that Gag initially binds to non-raft regions of the PM and subsequently either laterally associates with raft like domains or induces their formation (*Ono and Freed, 2001*; *Mercredi et al., 2016*).

Collectively, our study shows that PI(4,5)P$_2$ at the PM is crucial throughout the entire HIV-1 assembly process and suggests a dynamic equilibrium of Gag-lipid interactions. Furthermore, it establishes an experimental system that permits synchronized induction of HIV-1 assembly by targeted modulation of Gag PM targeting.

## Materials and methods

### Chemicals, reagents, plasmids

All chemicals and reagents were purchased from commercial sources unless otherwise noted. rCD1 was synthesized according to previously described procedures (*Feng et al., 2014*). FK506 was purchased from LC laboratories (Woburn, MA, USA).

Plasmid pCHIV, expressing all HIV-1 NL4-3 proteins except for Nef under the control of a CMV promotor and its derivatives pCHIV$^{EGFP}$, pCHIV$^{SNAP}$ and pCHIV$^{CLIP}$ were described previously (*Lampe et al., 2007*; *Eckhardt et al., 2011*; *Hanne et al., 2016*). Derivatives pCHIV(Δvpu) and pCHIV$^{EGFP}$(Δvpu) carry a deletion within the *vpu* coding region (bp 3–120), resulting in truncation of 40 amino acids followed by a frameshift. The design of pCHIV(Δ8-126 SR), kindly provided by Martin Obr, was based on the previously published MA deletion Δ8-126 (*Reil et al., 1998*), in which the globular domain of MA was replaced by two foreign amino acids (Ser-Arg). A SNAP-tagged variant, pCHIV$^{SNAP}$(Δ8-126 SR), was generated by gene synthesis (Thermo Fisher Scientific Geneart, Regensburg Germany) based on pCHIV$^{SNAP}$. The EGFP-tagged variant was generated by exchanging a ClaI fragment comprising the SNAP-tag coding region from pCHIV$^{SNAP}$(Δ8-126 SR) by a ClaI fragment of pCHIV$^{EGFP}$ (*Lampe et al., 2007*) that comprises the GFP coding region. In the resulting plasmid pCHIV$^{EGFP}$(Δ8-126 SR) the globular domain of MA is replaced by two foreign amino acids (Ser-Arg), which are connected to EGFP by a flexible linker. EGFP and CA are, similar to pCHIV$^{EGFP}$, connected via the HIV-1 protease cleavage site. Plasmid sequence information of the pCHIV(Δ8-126 SR) constructs are provided as supplementary files. For the generation of pGag-EGFP(G2A), Gag was PCR amplified from pGag-EGFP (*Hermida-Matsumoto and Resh, 2000*) using the primer 5′-CCC AAG CTT ATG GCT GCG AGA GCG TCG-3′, which contains a HindIII restriction site and introduces the G2A mutation and the primer 5′-CGG GAT CCC CTT GTG ACG AGG GGT CGC-3′, containing a BamHI cleavage site. The PCR product was digested using BamHI and HindIII restriction endonucleases and ligated with a BamHI/HindIII cleavage product of pGag-EGFP. pNL4-3 was described previously (*Adachi et al., 1986*). pNL4-3 16EK/29/31KE (*Joshi et al., 2009*) and pNL4-3 25/26KT (*Freed et al., 1994*) mutants were kindly provided by Eric Freed (NCI, Frederick). pEGFP-PLCδ-PH was described previously (*Várnai and Balla, 1998*). The reversible chemical dimerizer system consisting of plasmids pLCK-ECFP-SNAPf and pmRFP-FKBP-5Ptase were described previously (*Feng et al., 2014*; *Varnai et al., 2006*). The EGFP-tagged version pEGFP-FKBP-5Ptase was kindly provided by Martina Schifferer. Plasmids pWPI_Puro and pWPI_BLR were obtained from Oliver Fackler (*Trotard et al., 2015*). For generation of the lentiviral expression vectors pWPI-Puro-LCK-ECFP-SNAPf and pWPI-BLR-mRFP-FKBP-5Ptase, the Anchor (LCK-ECFP-SNAPf) and the Enzyme (mRFP-FKBP-5Ptase) coding regions were PCR-amplified from pLCK-ECFP-SNAPf and pmRFP-FKBP-5Ptase using the primers 5′-atcgaGGCGCGCCATGGGCTGCGTGTGCAG-3′ with 5′-atcgaACTAGTTTAATTAACCTCGAGTTTAAACGC-3′ for amplification of LCK-ECFP-SNAPf and 5′-gataaGGCGCGCCATGGCCTCCTCCGAGGA-3′ with 5′-gctacACTAGTTCAAGAAACGGAGGCGATG-3′ for amplification of mRFP-FKBP-5Ptase, introducing AscI and SpeI restriction sites at the 5′ and 3′ end, respectively. The fragments were digested using AscI and SpeI and subcloned into the vectors pWPI-Puro (Anchor) and pWPI-BLR (Enzyme) via AscI and SpeI restriction sites. Vesicular stomatitis virus G protein expression plasmid pMD2.G (Addgene#12260) and the lentiviral packaging plasmid psPAX2 (Addgene#12259) were a gift from Didier Trono (EPFL, Lausanne, Switzerland). pAdVantage was obtained from Promega (Mannheim, Germany).

## Generation of the HeLa$_{rCDS}$ cell line

The cell line HeLa$_{rCDS}$ was generated using a lentiviral vector system. 293T cells seeded in 6-well plates were co-transfected with psPax2, pMD2.G, pAdvantage and either pWPI-Puro-LCK-ECFP-SNAPf or pWPI-BLR-mRFP-FKBP-5Ptase at a molar ratio of 0.27: 0.25: 0.13: 0.35 using Turbofect (Thermo Scientific, Waltham, USA) according to the manufacturer's instructions. Tissue culture supernatants were harvested at 48 hpt, cleared by centrifugation and sterile filtered (0,45 μm pore size). 100 μl of both supernatants were mixed and used to transduce HeLa Kyoto cells in 6-well plates. Two days after transduction the growth medium was replaced by growth medium containing 2 μg/ml Puromycin (Merck Millipore, Billerica, USA) and 5 μg/ml Blasticidin (Thermo Scientific, Waltham, USA) for selection. Cells were passaged in selection medium and expression of the Anchor and Enzyme was monitored by spinning disc confocal microscopy and flow cytometry. Three weeks post transduction 100% of the HeLa$_{rCDS}$ cell pool expressed the Enzyme, while 93% expressed the Anchor.

## Cell culture and transfection

HeLa Kyoto (RRID:CVCL_1922), HEK293T (RRID: CVCL_0063) and TZM-bl reporter cells (RRID:CVCL_B478) (*Wei et al., 2002*) were cultured at 37°C and 5% $CO_2$ in Dulbecco's modified Eagle's medium (DMEM; Invitrogen) supplemented with 10% fetal calf serum (FCS; Biochrom), 100 U/ml penicillin and 100 µg/ml streptomycin. Medium for cultivation of HeLa$_{rCDS}$ cells additionally contained 2 µg/ml Puromycin and 5 µg/ml Blasticidin. Cell line identity has been authenticated using STR profiling (Promega PowerPlex 21 Kit; carried out by Eurofins Genomics, Ebersberg, Germany). Cell lines were grown from mycoplasma free liquid nitrogen stocks. Passaged cells in culture in the lab are monitored regularly (every 4 months) for mycoplasma contamination using the MycoAlert mycoplasma detection kit (Lonze Rockland, USA) and cell lines used here were contamination free.

For microscopy experiments, cells were seeded on slides within eight-well Lab-Tek chambered coverglass systems (Thermo Scientific, Waltham, USA). At about 50% confluence, cells were transfected using Turbofect (Thermo Scientific, Waltham, USA) according to the manufacturer's instructions. 0.5 µg DNA was transfected per well with pLCK-ECFP-SNAP, pmRFP-FKBP-5Ptase or pEGFP-FKBP-5Ptase, as indicated, and pCHIV derivatives (molar ratio of 1.8:1:0.5). Tagged pCHIV derivatives were transfected in equimolar ratio with their non-labeled counterpart resulting in a ratio of 1.8:1:0.25:0.25. In the case of pNL-43 variants a molar ratio of 1.8:1:0.43 (pLCK-ECFP-SNAP: pmRFP-FKBP-5Ptase: pNL4-3) was applied. pGag-EGFP(G2A) was used at a molar ratio of 2:1 with pCHIV and DNA amount of HIV-derivatives was increased 3.6-fold, while pLCK-ECFP-SNAP and pmRFP-FKBP-5Ptase quantities were kept constant, resulting in a total DNA amount of 0.76 µg/well and a molar ratio of pLCK-ECFP-SNAP:pmRFP-FKBP-5Ptase:pCHIV:pGag-EGFP(G2A) 1.8:1:0.9:1.8. If pEGFP-PLCδ-PH was used, total DNA amount was reduced to 0.41 µg/well, while pLCK-ECFP-SNAP and pmRFP-FKBP-5Ptase quantities were kept constant, resulting in a molar ratio for pLCK-ECFP-SNAP: pmRFP-FKBP-5Ptase: pEGFP-PLCδ-PH of 1.8: 1: 0.05.

At 4 hr post transfection (hpt) the transfection mixture was replaced by imaging medium (DMEM high glucose w/o phenol red w/o glutamine supplemented with 10% FCS, 4 mM GlutaMAX, 2 mM sodium pyruvate, 20 mM HEPES pH 7.4, 100 U/ml penicillin and 100 µg/ml streptomycin).

## Preparation of live cell samples

In order to achieve PM PI(4,5)$P_2$ depletion prior to Gag accumulation, 1 µM rCD1 or 1% dimethyl sulfoxide (DMSO) vector control was added to the imaging medium at 4 hpt. During image acquisition (performed at 22 hpt), samples were additionally treated with 1 µM FK506 to achieve PM PI(4,5)$P_2$ reconstitution or with 1% DMSO, as indicated. In order to achieve PM PI(4,5)$P_2$ depletion after Gag assembly at the PM, transfected samples were treated with 1 µM rCD1/1% DMSO at 22 hpt for a period of 90 min during live cell imaging. Subsequently 1 µM FK506 or 1% DMSO, as indicated, was added for an additional period of 90 min to achieve PM PI(4,5)$P_2$ reconstitution. The medium was not exchanged between treatments. For Cycloheximide (CHX) treatment, cells were incubated with 10 µg/ml CHX (Sigma-Aldrich, St. Louis, USA).

## CLIP-labeling and immunofluorescence staining

For STED imaging HIV-1 Gag was detected via a CLIP-tag (expressed from pCHIV$^{CLIP}$[*Hanne et al., 2016*]). Living cells were stained with Atto 590 BC-CLIP (kindly provided by Janina Hanne) in imaging medium for 30 min. Cells were washed twice with imaging medium, incubated for further 30 min at 37°C 5% $CO_2$ in imaging medium, washed with phosphate buffered saline (PBS) and fixed for immunostaining.

For immunostaining, cells were fixed at 24 hpt for 15 min with 4% PFA in PBS. For samples expressing the pNL4-3 mutants 16EK/29/31KE or 25/26KT fixation was prolonged to 90 min for biosafety reasons. Samples were washed with PBS, permeabilized with 0.1% Triton-X100 in PBS for 5 min and blocked for 30 min with 2% BSA in PBS. Permeabilization was omitted in case of Env labeling. Cells were incubated with the indicated primary antibody in 2% BSA in PBS for 2 hr, washed with PBS and incubated with the respective secondary antibody or Fab in 2% BSA in PBS for 1 hr. Antibody/Fab combinations were as follows: MA: polyclonal rabbit antiserum (in house) 1:500 / anti rabbit Alexa Fluor 647 1:500 (Thermo Fisher Scientific Cat# A-21244 RRID:AB_2535812), Env: monoclonal anti-gp120 antibody 2G12 1:50 (Polymun Scientific Cat#AB002 RRID:AB_2661842) / Fab Fragment Goat Anti-Human IgG (H+L) (Jackson ImmunoResearch Labs Cat# 109-007-003 RRID:AB_

2337555) coupled to Abberior STAR RED NHS (Abberior Instruments GmbH, Göttingen, Germany) 1:50, Tetherin: monoclonal anti-CD317 antibody 26F8 1:200 (Thermo Fisher Scientific Cat# 16-3179-82, RRID:AB_1518775) / anti mouse Alexa 647 1:500 (Thermo Fisher Scientific Cat# A-21236 RRID: AB_2535805). Finally, cells were washed and kept in PBS.

## Microscopy

Spinning disk confocal (SDC) imaging was performed at a PerkinElmer UltraVIEW VoX SDC microscope (Perkin Elmer, Waltham, USA) using a 60x Apo TIRF (NA 1.49) oil immersion objective and Hamamatsu C9100-23B EM-CCD camera. Stacks were acquired with a z-spacing of 500 nm. Live-cell imaging was performed at 37°C, 5% $CO_2$, 40% humidity using multiposition imaging with an automated stage and the Perfect Focus System (Nikon, Tokio, Japan) for automated focusing at each time point with a time-resolution of 6 s - 5 min/frame.

Total internal reflection fluorescence (TIRF) live cell imaging was performed at objective type TIRF setup (Visitron Systems, Puchheim, Germany) based on a Zeiss Axiovert 200M fluorescence microscope equipped with a 100 x Zeiss Alpha Plan Apochromat (NA 1.46) oil immersion objective and Hamamatsu EM-CCD 9100–50 camera. The TIRF angle was manually controlled. Live-cell imaging was performed at 37°C 5% $CO_2$, 40% humidity with a time-resolution of 5 s/frame.

Stimulated emission depletion (STED) imaging was performed at a $\lambda$ = 775 nm STED system (Abberior Instruments GmbH, Göttingen, Germany), using a 100 x Olympus UPlanSApo (NA 1.4) oil immersion objective with 590 and 640 nm excitation laser lines at room temperature. Nominal STED laser power was set to ~60% of the maximal power of 1200 mW with 10 µs pixel dwell time and 15 nm pixel size.

## Image representation

Representative still images or single frames of image sequences were chosen. Super resolution STED images were deconvolved with a Lorentzian function (full width half maximum = 60 nm) using the software Imspector (Abberior Instruments GmbH, Göttingen, Germany). For all images shown, the camera offset value was subtracted and the contrast and brightness were adapted for optimal display of the image. To eliminate background noise, a 0.5 px median filter was applied to all SDC and TIRF images. Images are shown in greyscale or pseudo colors. In the latter case, the Fire and Green Fire Blue lookup tables (LUTs) were used for SDC and TIRF images, respectively, while different channels of super resolution STED images are shown with the LUTs Red Hot (referred to as 'red') and Green Fire Blue (referred to as 'cyan').

## Image analyses

### Particle analysis of SDC microscopy data

Out of all cells imaged, those which showed the respective Enzyme translocation and maintained a healthy phenotype throughout the whole imaging process were selected for further analysis. Cells that showed clear signs of apoptosis, e.g. severe shrinking, were excluded from the analysis.

Particle analysis of SDC microscopy data was done in FIJI (RRID:SCR_002285) (*Schindelin et al., 2012*) on single slices of the ventral PM or on maximum intensity projections of four slices located in the middle of the cell, as indicated. All parameters for image processing were kept constant when comparing different data sets. First the camera offset value was subtracted from all images. To remove the image background and prepare the images for automatic thresholding, single images - or image sequences in case of time lapse imaging - were converted to 8-bit images and background was subtracted using a rolling ball (radius = 2 px). The objects of interest were automatically thresholded using the Niblack local thresholding method (*Niblack, 1986*) with the following parameters: radius = 10, parameter 1 = 0 and parameter 2=-25. A 0.5 px median filter was applied to all images. The region of interest was selected manually by drawing the outline of the respective cell. Finally, the size and number of particles in each cell for every still image or every frame of a timelapse sequence was determined using FIJI's Analyze Particles function with the following parameters: size = 2-infinity $px^2$, circularity = 0–1. Values were exported and further analyzed in Excel (Microsoft, Redmond, USA) or GraphPad Prism (GraphPad Software, Inc., La Jolla, USA; RRID:SCR_002798).

For still images, the number of particles detected at the ventral PM was divided by membrane area and plotted as number of Gag clusters/1000 $\mu m^2$.

In time lapse experiments obtained at the ventral cell membrane of cells treated with rCD1 or DMSO at 4 hpt (before Gag accumulation), the number of detected particles in the 3–10 frames before addition of FK506 or DMSO was averaged and subtracted from all data points to exclude Gag clusters already present before addition of the respective compound. Values were normalized for membrane area and plotted as number of Gag clusters/1000 $\mu m^2$ over time.

For time lapse experiments imaged in a central section of cells treated with rCD1 or DMSO at 22 hpt (after Gag assembly at the PM), the number of particles in the 3–10 frames before addition of rCD1 or DMSO was averaged and set to 100%. All values were normalized accordingly and plotted as relative number of Gag clusters over time.

## Particle analysis of TIRF microscopy data

Particle analysis of TIRF microscopy data was done in FIJI (RRID:SCR_002285) (*Schindelin et al., 2012*) in an overall similar manner as described for particle analysis of SDC microscopy data. The following parameters were changed in order to achieve optimal particle identification in TIRF image sequences: radius of rolling ball was 20 px, radius for Niblack automated thresholding was 10, parameter 2 was −10, a 1 px median filter was applied, the size in FIJI's Analyze Particles function was set to 3-infinity $px^2$.

Values were exported and further analyzed in Excel (Microsoft, Redmond, USA) ore GraphPad Prism (GraphPad Software, Inc., La Jolla, USA; RRID:SCR_002798). All values obtained were divided by membrane area. For temporal alignment of movie sequences, the frame in which ~100 particles/1000 $\mu m^2$ were detected was defined as t = 0. Finally, values were plotted as number of Gag clusters/1000 $\mu m^2$ over time.

## Cluster intensity analysis

Cluster intensity analysis was done in FIJI (RRID:SCR_002285) (*Schindelin et al., 2012*). In contrast to the particle analysis, sum intensity projections of four slices acquired with 500 nm spacing in the middle of the cell were used. Clusters were thresholded as described above (see Particle analysis of SDC microscopy data). The mean fluorescence intensity in clusters was quantified using the thresholded sum projected images and plotted over time.

## STED line profiles of assembly sites

Images were linearly deconvolved with a Lorentzian function (full width half maximum = 60 nm) using the software Inspector (Abberior Instruments GmbH, Göttingen, Germany). Line profiles of selected assembly sites were generated manually in FIJI (RRID:SCR_002285) (*Schindelin et al., 2012*). The intensity values in the Gag and Env channel were exported to Excel. To align the line profiles of different assembly sites, the Gag intensity peak of an assembly site was set to x = 0 nm and the corresponding Env intensity profile was adjusted accordingly. Intensity values were exported to GraphPad Prism (GraphPad Software, Inc., La Jolla, USA; RRID:SCR_002798), normalized (smallest value = 0, highest value = 100) and the average normalized fluorescence intensities ± SEM were plotted.

## Single assembly site kinetics

Time courses of native or induced Gag assembly at individual sites at the ventral PM were extracted from single plane TIRF microscopy time lapse sequences at a time resolution of 5 s/frame. Time courses of reinduced and native assembly at individual sites at the lateral PM were extracted from 3D volume time lapse (five slices, 0.5 $\mu m$ spacing) acquired by SDC microscopy at a time resolution of 6 s/frame.

The camera offset value was subtracted from all image sequences using FIJI (RRID:SCR_002285) (*Schindelin et al., 2012*) and image sequences were imported to Imaris 8 (Bitplane AG, Zurich, Switzerland). Spot detection and tracking was performed using Imaris' Spot detection module. Within this process the background was subtracted and the estimated diameter for spot detection was set to 500 nm (TIRF) or 700 nm (SDC). The quality parameter for spot detection was in the range of 35–250 (TIRF) or 75–200 (SDC), depending on the dataset. Tracking was performed using the Autoregressive Motion algorithm, assuming a maximum distance between frames of 500 nm (TIRF) or 700–1000 nm (SDC), allowing for a maximum gap size of 1 (TIRF) or 4 (SDC) and a track duration above

300 s. Filling gaps was disabled. Out of the detected spots, those which increased in mean intensity and reached a plateau phase were selected for further analysis. The mean intensity values over time were exported to Excel, temporally aligned (the beginning of each track was set to t = 0) and normalized (smallest value = 0, highest value = 100). The average normalized fluorescent values (a.u.) including standard deviation over time were plotted. In order to calculate assembly rate constants and half-times, single exponential fits to the data were performed using GraphPad Prism software (GraphPad Software, Inc., La Jolla, USA; RRID:SCR_002798). The frequency distribution of k-values was plotted after binning with a bin width of 0.0005 $s^{-1}$ and 120 s, respectively.

## Release and infectivity assays

HeLa$_{rCDS}$ cells were seeded in 6-well plates and transfected with 2 µg pNL4-3 at about 50% confluence using Turbofect (Thermo Scientific, Waltham, USA) according to the manufacturer's instructions. At 5 hpt cells were treated with 1 ml growth medium containing 1 µM rCD1 (induced/inhibited assembly) or equivalent amounts of DMSO (native assembly) at 5 hpt. At 22 hpt the tissue culture supernatant was harvested ('0 hr' sample) and cleared by centrifugation at 1000 g for 10 min. The cells were washed with PBS twice and virus attached to the cell surface was inactivated by acid wash (40 mM citric acid, 135 mM NaCl, 10 mM KCl, pH 3.0) for 1 min followed by a washing step with pre-warmed medium. Subsequently, 1 ml of fresh, pre-warmed growth medium containing 1 µM rCD1 (inhibited assembly), 1 µM FK506 (induced assembly) or DMSO solvent control (native assembly) was added to the cells and incubation was continued for 2 hr. Supernatants were harvested again ('2 hr' samples) and cleared by centrifugation at 1000 g for 10 min. Cells were processed for immunofluorescence and flow cytometry to control for transfection efficiency and Enzyme localization.

Relative virus infectivity was assessed using TZM-bl reporter cells. 5*10³ TZM-bl cells/well were plated in 96-well plates and infections with serial 2-fold dilutions of the cleared supernatants obtained from virus producing cells were carried out on the following day. At 48 hpi (hours post infection), infected cells were lysed and luciferase activity was measured using the Steady-Glo-Assay (Promega, Madison, USA) according to the manufacturer's instructions. Relative infectivity of supernatants was calculated from the linear range of the titration curves and normalized to the amount of virus present in the supernatants as assessed by SG-PERT (SYBR Green based Product Enhanced Reverse Transcriptase) assay (*Pizzato et al., 2009*).

## Statistical analysis

Data analysis was performed using GraphPad Prism (GraphPad Software, Inc., La Jolla, USA; RRID: SCR_002798). Values are expressed as mean ±SEM or mean ±SD, as indicated. Statistical significance of the data presented was assessed with the two-tailed unpaired Student's t-test or the nonparametric Mann-Whitney U test, as indicated. Mann-Whitney U test was applied for statistical analysis of the data, which did not follow a Gaussian distribution (determined by inspection of Box and Whiskers graphs and Histograms of frequency distributions). Values of p<0.05 were considered significant.

## Supplemental materials

*Supplementary files 1*, *2,* and *3*. Files pCHIV(Δ8-126 SR).txt, pCHIV$^{GFP}$(Δ8-126 SR).txt and pCHIV$^{S-NAP}$(Δ8-126 SR).txt contain the plasmid sequences of the respective constructs.

*Figure 1—figure supplement 1* shows reversible PI(4,5)P$_2$ depletion from the PM by the rCDS. *Figure 2—figure supplement 1* shows that the MA deletion mutant Δ8-126SR is not responsive to PI(4,5)P$_2$ depletion. *Figure 3—figure supplement 1* shows the relative frequency distributions of assembly rates of native and induced Gag assembly at the ventral PM. *Figure 5—figure supplement 1* shows the individual data sets determined for virus release and infectivity used for calculation of the normalized numbers shown in *Figure 5*. *Figure 6—figure supplement 1* shows that addition of DMSO does not affect assembled Gag clusters and that the MA deletion mutant Δ8-126SR does not respond to PI(4,5)P$_2$ depletion and rescue. *Figure 6—figure supplement 2* shows that virus release or impairment of new particle formation do not significantly contribute to the loss of Gag clusters from the PM upon PM PI(4,5)P$_2$ depletion. *Figure 6—figure supplement 3* shows that the Gag mutant G2A is not responsive to PI(4,5)P$_2$ rescue following PI(4,5)P$_2$ depletion. *Figure 6—figure supplement 4* shows the assembly kinetics of reinduced and native assembly at the lateral PM.

*Video 1* shows reversible PI(4,5)P$_2$ depletion from the PM by the rCDS, supporting *Figure 1—figure supplement 1*. *Video 2* shows rCDS-induced Gag assembly and *Video 3* shows the respective DMSO control, together supporting *Figure 2*. *Videos 4* and *5* show TIRF microscopy time-lapse of native and induced assembly, respectively, supporting *Figure 3*. *Video 6* show loss of Gag clusters upon PI(4,5)P$_2$ depletion and subsequent reinduction of Gag assembly by the rCDS and *Video 7* shows the respective DMSO control, together supporting *Figure 6*. *Video 8* shows reinduced assembly acquired with high timeresolution spinning disc confocal microscopy, supporting *Figure 6—figure supplement 4*.

## Acknowledgements

We are grateful to Martina Schifferer (EMBL Heidelberg), Alexander Jordan (University Hospital Heidelberg), Eric Freed (NCI Frederick), Martin Obr (University Hospital Heidelberg), Didier Trono (EPFL Lausanne) and Oliver Fackler (University Hospital Heidelberg) for providing plasmids, Janina Hanne (DKFZ and University Hospital, Heidelberg) for providing Atto590-CLIP and Suihan Feng (EMBL Heidelberg) for synthesizing rCD1. We thank Mathew Betts (MathClinic at Bioquant Heidelberg) for advice regarding statistical analyses and Susann Kummer (University Hospital, Heidelberg) for advice regarding STED microscopy.

## Additional information

### Competing interests

CS: C. Schultz is a shareholder of the company SiChem, which distributes rCDS. The other authors declare that no competing interests exist.

### Funding

| Funder | Grant reference number | Author |
|---|---|---|
| Deutsche Forschungsgemeinschaft | TRR 83 project 14 | Hans-Georg Kräusslich |
| Deutsche Forschungsgemeinschaft | TRR 83 project 2 | Carsten Schultz |
| Deutsche Forschungsgemeinschaft | SFB 1129 project 5 | Hans-Georg Kräusslich |
| Deutsche Forschungsgemeinschaft | SFB 1129 project 6 | Barbara Müller |
| Deutsche Forschungsgemeinschaft | Excellence Cluster CellNetworks Exc81 | Barbara Müller Hans-Georg Kräusslich |
| Deutsches Zentrum für Infektionsforschung | Project 7.5 TTU HIV | Hans-Georg Kräusslich |

The funders had no role in study design, data collection and interpretation, or the decision to submit the work for publication.

### Author contributions

FM, Conceptualization, Formal analysis, Investigation, Visualization, Writing—original draft, Writing—review and editing; VL, Conceptualization, Supervision, Validation, Investigation, Writing—review and editing; BM, Conceptualization, Resources, Supervision, Funding acquisition, Writing—review and editing; CS, Resources, Funding acquisition, Writing—review and editing; H-GK, Conceptualization, Supervision, Funding acquisition, Writing—original draft, Project administration, Writing—review and editing

### Author ORCIDs

Frauke Mücksch, http://orcid.org/0000-0002-0132-5101
Barbara Müller, http://orcid.org/0000-0001-5726-5585
Hans-Georg Kräusslich, http://orcid.org/0000-0002-8756-329X

## Additional files

**Supplementary files**
• Supplementary file 1. Plasmid sequence pCHIV(d8-126 SR).

• Supplementary file 2. Plasmid sequence pCHIVGFP(d8-126 SR).

• Supplementary file 3. Plasmid sequence pCHIVSNAP(d8-126 SR).

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
