## [Decision Letter]

Thank you for submitting your article "Synchronized HIV assembly by tunable PIP2 changes reveals PIP2 requirement for stable Gag anchoring" for consideration by *eLife*. Your article has been favorably evaluated by Wenhui Li (Senior Editor) and three reviewers, one of whom, Wesley I Sundquist (Reviewer #1), is a member of our Board of Reviewing Editors. Eric Freed (Reviewer #3) has agreed to reveal his identity.

The reviewers have discussed the reviews with one another and the Reviewing Editor has drafted this decision to help you prepare a revised submission.

Summary:

The authors use a chemical biology system that can induce rapid PI(4,5)P_2_ depletion, followed by rapid PI(4,5)P_2_ restoration, to examine the PIP2-dependence of HIV-1 Gag protein. They make three major claims:

1) PIP2 is required for HIV Gag membrane association/assembly (previously known, but nicely demonstrated).

2) PIP2 is also required to maintain preassembled Gag clusters at the plasma membrane (unexpected, and implying that Gag-lipid interactions are apparently more dynamic than had previously been appreciated).

3) The system permits synchronization of particle release (because the rates of Gag molecule addition at each assembly site are the same under native vs. PIP2 restoration conditions, but the overall frequency of assembly events per unit time is higher when PI(2)P is restored).

Overall, the system provides a strong experimental support for the role of PI(4,5)P_2_ in Gag PM binding without a concern of indirect effects of long-term PI(4,5)P_2_ depletion. The application is clever, the different assembly reactions are generally well characterized, and the unexpected requirement for PIP2 in maintaining the preassembled Gag lattice is a nice advance, but there are several issues that should be addressed before the manuscript is acceptable for publication in *eLife*.

Essential revisions:

1) The authors show that reassembly is similar to native assembly by a number of criteria (size, morphology, composition). However, the ultimate test of similarity is viral infectivity, and the generation of infectious virus is also required to realize the full potential of the synchronization reaction. It is therefore important that the authors test whether synchronized assembly upon PIP2 depletion/restoration produces virions that are natively infectious. Along the same lines, the authors present the surprising result that when Gag clusters are lost from the plasma membrane and then induced to re-assemble, the spatial distribution of Env is unaffected; i.e., the spatial distribution of Gag and Env at "native" vs. "induced" assembly sites does not appear to differ. If correct, this result would have implications for how we think about Env recruitment to Gag assembly sites. It is therefore important to corroborate this result by measuring Env incorporation into virions produced in the native vs. induced setting.

2) The data shown in Figure 5 are presented to demonstrate that PIP2 depletion results in loss of preassembled Gag clusters. Appropriate controls are included (DMSO addition and delta8-126SR – although the graphs in 5A and 5B should also show the quantified data for the delta8-126SR constructs). Importantly, however, steady-state changes in Gag cluster numbers under each condition could reflect changes in: A) The rate of new Gag cluster formation, B) The rate of preformed Gag cluster dissociation into the cytoplasm (and through dispersion along the membrane – can this be ruled out?, If not, could dispersal of Gag clusters in the membrane rather than loss from the membrane explain the more rapid accumulation of Gag clusters upon dissociation and reinduction (Figure 5)?), and C) The rate of Gag cluster loss through extracellular budding. Can the authors account for possible changes in the rates of each of these different processes, and thereby convincingly say that most of the loss in cluster numbers for wt Gag upon treatment with rCD1 results primarily from dissociation of preformed Gag clusters (and not from changes in rates of cluster initiation or particle budding?). Are there complementary ways to look more directly at preformed Gag cluster dissociation; for example by following the fates of photoconverted, preassembled Gag clusters to rule out contributions from new Gag cluster formation, by blocking virus budding to rule out contributions from extracellular particle release, by analyzing the fates of individual clusters quantitatively rather than reporting bulk net changes and/or by adding CHX instead of rCD1 to WT Gag-expressing cells and monitoring the decrease of Gag clusters over time (if the decrease is similar to that observed with rCD1, the effect observed with rCD1 is not due to detachment of Gag lattice but rather due to the combination of the lack of new Gag binding to the PM and release of pre-existing Gag lattice as virus particles to the extracellular space)?

---

## [Author Response]

*Essential revisions:*

*1) The authors show that reassembly is similar to native assembly by a number of criteria (size, morphology, composition). However, the ultimate test of similarity is viral infectivity, and the generation of infectious virus is also required to realize the full potential of the synchronization reaction. It is therefore important that the authors test whether synchronized assembly upon PIP2 depletion/restoration produces virions that are natively infectious. Along the same lines, the authors present the surprising result that when Gag clusters are lost from the plasma membrane and then induced to re-assemble, the spatial distribution of Env is unaffected; i.e., the spatial distribution of Gag and Env at "native" vs. "induced" assembly sites does not appear to differ. If correct, this result would have implications for how we think about Env recruitment to Gag assembly sites. It is therefore important to corroborate this result by measuring Env incorporation into virions produced in the native vs. induced setting.*

We agree with the reviewer that demonstrating virus infectivity after induction, which also depends on Env and RNA recruitment to the induced assembly site, is an important experiment providing validation for this approach. We had already attempted to demonstrate this using the co-transfection approach employed for the other experiments. However, low levels of virus obtained in short time periods after induction, combined with background resulting from cells expressing HIV in the absence of the complete rCDS hampered reliable interpretation of results, which we therefore decided not to include in the original manuscript.

We have now established and characterized a stable HeLa based cell line expressing both rCDS components. Employing this novel tool for native and induced virus production, we could indeed confirm that induced Gag targeting leads to the production of viruses with a relative infectivity of ca. 65% compared to control particles. These data are included as new Figure 5 with supplements and are discussed in the revised manuscript text. We believe that these results also confirm that sufficient Env molecules are incorporated into particles obtained by induced assembly. Furthermore, our super-resolution microscopy analyses show similar distributions of Gag and Env at induced and native assembly sites, indicating that induced assembly leads to similar Env recruitment as native assembly.

*2) The data shown in Figure 5 are presented to demonstrate that PIP2 depletion results in loss of preassembled Gag clusters. Appropriate controls are included (DMSO addition and delta8-126SR – although the graphs in 5A and 5B should also show the quantified data for the delta8-126SR constructs).*

As requested by the reviewer, we have quantified the data for the delta8-126SR mutant. In order to avoid overcrowding of the main figure, we have added the data as new panels D and E to the revised Figure 6—figure supplement 1.

*Importantly, however, steady-state changes in Gag cluster numbers under each condition could reflect changes in: A) The rate of new Gag cluster formation, B) The rate of preformed Gag cluster dissociation into the cytoplasm (and through dispersion along the membrane – can this be ruled out?, If not, could dispersal of Gag clusters in the membrane rather than loss from the membrane explain the more rapid accumulation of Gag clusters upon dissociation and reinduction (Figure 5)?), and C) The rate of Gag cluster loss through extracellular budding. Can the authors account for possible changes in the rates of each of these different processes, and thereby convincingly say that most of the loss in cluster numbers for wt Gag upon treatment with rCD1 results primarily from dissociation of preformed Gag clusters (and not from changes in rates of cluster initiation or particle budding?). Are there complementary ways to look more directly at preformed Gag cluster dissociation; for example by following the fates of photoconverted, preassembled Gag clusters to rule out contributions from new Gag cluster formation, by blocking virus budding to rule out contributions from extracellular particle release, by analyzing the fates of individual clusters quantitatively rather than reporting bulk net changes and/or by adding CHX instead of rCD1 to WT Gag-expressing cells and monitoring the decrease of Gag clusters over time (if the decrease is similar to that observed with rCD1, the effect observed with rCD1 is not due to detachment of Gag lattice but rather due to the combination of the lack of new Gag binding to the PM and release of pre-existing Gag lattice as virus particles to the extracellular space)?*

We of course agree with the reviewers that the number of Gag clusters at steady state conditions is determined by the rate of formation of assembly sites and the rate of their disappearance either into the extracellular space or back into the cytoplasm. We have addressed the reviewers’ comments as follows:

A) The rate of new cluster formation – in response to this comment, we have performed experiments in which new protein synthesis was blocked by the addition of CHX as suggested by the reviewers. We have then monitored changes in the number of Gag clusters over time, where continuous depletion and repletion of assembly site even under native conditions should lead to loss of PM Gag clusters. This was not the case, however. As shown in the new Figure 6—figure supplement 2, blocking protein expression did not change the number of Gag clusters at the PM indicating their stable PM association in the absence of PI(4,5)P_2_ depletion.

B) The dispersal of Gag clusters in the PM – dispersal of a significant fraction of Gag clusters within the PM should lead to a significant increase in the PM staining of Gag. This was not observed, however (see e.g. Video 6). We do not consider membrane flotation experiments to be conclusive in this regard as our main conclusion is that Gag is lost from the PM, but not necessarily from any cellular membrane. Thus, Gag removed from the PM may at least partly remain associated with cytoplasmic membranes. One would predict that Gag clusters removed from the membrane may not dissociate completely, but remain in an oligomeric state, but this is difficult to detect over the large background of cytoplasmic Gag molecules.

C) Rate of Gag cluster loss through extracellular budding – to address this question, we performed the same experiment under conditions that impair particle release from the cell surface. For this, we used a *vpu* deletion mutant of pCHIV. Vpu downregulates plasma membrane levels of the host cell restriction factor tetherin, which retains fully budded HIV-1 particles at the cell surface. In the absence of Vpu HIV-1 release is thus inhibited. Control experiments confirmed that tetherin was depleted from the PM in the presence of Vpu under our conditions, and depletion was not observed with the Vpu(-) virus (new Figure 6—figure supplement 2). Further controls showed that the localization of tetherin was not affected by PI(4,5)P_2_ depletion (new Figure 6—figure supplement 2). Live cell imaging of Gag assemblies in the absence of Vpu following PI(4,5)P_2_ depletion revealed that Gag clusters were lost from the PM with similar efficiency compared to experiments performed using the Vpu-expressing construct (Figure 5—figure supplement 2E and F). These observations indicate that the loss of Gag clusters upon PI(4,5)P_2_ depletion is not reflecting a change in the rate of Gag cluster loss through extracellular buding.

Results from all these additional experiments are discussed in the subsections “PI(4,5)P_2_ is required to retain the partially assembled Gag lattice at the PM” and “HIV-1 Gag assembly sites can be reconstituted by rescue of PM PI(4,5)P_2_”.